# Antibody function predicts viral control in newborn monkeys immunised with an influenza virus HA stem nanoparticle

Kali F. Crofts[1,4], Beth C. Holbrook [1], Courtney L. Page [1], Rebecca A. Gillespie [2], Ralph B. D'Agostino Jr[3], Maya Sangesland[2], David A. Ornelles[1], Masaru Kanekiyo [2] & Martha A. Alexander-Miller [1] ✉

The lack of an approved influenza vaccine for infants <6 months, coupled with the requirement for annual updates of current vaccines, warrants the development of a universal vaccine that can confer protection in young infants. Here we test the ability of a ferritin nanoparticle universal influenza vaccine (H1ssF) containing the stem region of hemagglutinin (HA) adjuvanted with AddaVax to elicit responses in newborn African green monkeys (AGM). Vaccinated newborns show robust HA stem-specific IgG responses but, despite the high antibody levels, viral load in the lung following H1N1 CaO9 challenge is variable among animals. Further analysis indicates that viral clearance is correlated with the presence of antibodies with neutralizing and antibody-dependent cellular phagocytosis activity. Our findings show that newborn AGM can generate functional HA stem-specific antibodies for viral clearance following vaccination with H1ssF+AddaVax and support further investigation of H1ssF as a universal vaccine for this vulnerable human population.

Influenza A virus (IAV) infections place a significant burden on children's health globally, particularly in infants under 6 months of age, who face an elevated risk of disease severity and mortality[1,2]. The heightened susceptibility of these naïve individuals is due to the altered responsiveness of the newborn immune system[3]. The immune landscape of newborns favours tolerance over strong inflammatory responses, which is proposed to allow for the establishment of the microbiome. Although necessary, the altered system results in challenges in generating a protective immune response following IAV infection and vaccination[4–7]. The lack of an approved seasonal influenza vaccine for infants under 6 months of age is due to the poor responses of infants to current formulations. Seroconversion rates against H1N1 strains may only reach 29–32% following two doses of vaccine in young infants[5,7]. Maternal antibodies (Abs) can often provide protection against IAV; however, in many cases, this protection is incomplete[8,9]. Further, maternal Ab wanes considerably before

newborns can receive the two doses of vaccine required to reach protective levels of immunity, creating a window of vulnerability[3]. Thus, the development of a vaccine that is safe and effective is crucial for the protection of this vulnerable population.

The newborn immune system exhibits alterations that span both innate and adaptive immunity, resulting in significant decreases in the production of high-affinity IgG during the first year of life[3,10]. This is, in part, a consequence of decreased plasma cell survival signals, decreased costimulatory molecule expression needed for generating T cell help, and diminished Ab affinity maturation[11]. Notably, there is a delay in the maturation of follicular dendritic cells (FDCs) that support the development and maintenance of germinal centre (GC) responses[11]. Collectively, these changes present substantial obstacles in eliciting protective Ab responses to current influenza vaccines.

An additional consideration in eliciting protective responses through vaccination in young infants is the potential for inhibitory

[1]Department of Microbiology and Immunology, Wake Forest University School of Medicine, Winston-Salem, NC, USA. [2]Vaccine Research Center, National Institute of Allergy and Infectious Diseases, National Institutes of Health, Bethesda, MD, USA. [3]Department of Biostatistics and Data Science, Wake Forest University School of Medicine, Winston-Salem, NC, USA. [4]Present address: Department of Surgery, Duke University School of Medicine, Durham, NC, USA. ✉e-mail: martha.alexander-miller@wakehealth.edu

effects imposed by maternal Ab. The impact of maternal Abs on the response to influenza vaccination in human infants remains unclear, primarily because vaccines are not administered to infants younger than six months[3]. The available data in human infants suggest maternal Ab can dampen infant responses to seasonal influenza vaccines, with higher maternal Ab associated with lower seroconversion rates[5]. Further, in a murine model, maternal Abs impeded differentiation of activated hemagglutinin (HA)-specific B cells into plasma cells, reducing the Ab response to vaccination[12]. Notably, the degree of inhibition appears to vary by vaccine platform, with mRNA-LNP vaccines mitigating the dampening effects seen with inactivated IAV vaccines[13].

Beyond the challenges associated with infant vaccination, current influenza vaccines offer minimal protection against drifted or pandemic strains of IAV. These vaccines predominantly generate neutralising antibodies (nAbs) specific to the hypervariable head region of the surface glycoprotein HA. The HA head is the main target of antigenic drift[14], necessitating annual updates to seasonal vaccines. To overcome this hurdle, there has been a focus on eliciting broadly reactive Abs that recognise conserved viral structures, e.g., the stem domain of the HA molecule, which is less susceptible to mutation due to its role in membrane fusion[15,16]. Although highly conserved, the HA stem is poorly immunogenic[17–21]. Recent advances in HA stem vaccine platforms have been successful in overcoming the immunodominance of the head region. Sequential immunisation with a series of chimeric influenza viruses with divergent HA head and conserved HA stem domains have proven effective in generating protective, broadly reactive responses to this region[15,20]. Other approaches have removed the HA head domain to overcome the issue of head immunodominance[22,23]. Yassine et al. showed the attachment of trimeric stem proteins (from H1N1 A/New Caledonia/20/1999) to a self-assembling ferritin (from *H. pylori*) nanoparticle (H1ssF) allowed for a stabilised trimeric conformation that induced broadly reactive Abs[22]. Specifically, animal studies evaluating H1ssF delivered in the presence of oil-in-water adjuvants have demonstrated strong serological responses against diverse group 1 IAV subtypes, resulting in Ab-mediated protection against heterosubtypic H5N1 infection in mice and ferrets[22] and generation of broadly nAbs in cynomolgus macaques[24]. Encouragingly, this vaccine has completed Phase 1 clinical trials in healthy adults (NCT03814720), exhibiting safe and durable nAb responses more than one year following vaccination[25]. Further, Andrews et al. reported H1ssF vaccinated individuals generated a plasmablast response and sustained memory B cell responses[26]. While these findings hold great promise for a universal vaccine in adults, the utility of this vaccine approach to promote broadly reactive, protective responses in the context of the newborn immune system has not been explored. This is critical given the numerous immune alterations present in the newborn.

In these studies, we utilise a newborn African green monkey (AGM) nonhuman primate (NHP) model. NHPs are the most relevant pre-clinical model due to their immunologic, developmental, and physiological similarities to humans[27,28]. This is coupled with a prolonged period of infancy compared to small animal models, allowing for assessment of a prime/boost immunisation in the context of a newborn immune system. We have previously shown that newborn AGMs are capable of producing HA stem-specific Abs following either IAV infection[29] or vaccination with R848-conjugated whole inactivated H1N1 virus[30]. Although these data showed newborn NHPs have the necessary repertoire to generate HA stem-specific Ab responses, these responses reflected the subdominance observed in adults. In this study, we determined whether we could drive a stronger HA stem-specific Ab response in newborns using the HA stem-only vaccine (H1ssF) and whether this response was capable of broad recognition of heterologous strains as well as protection.

Here, H1ssF adjuvanted with AddaVax is delivered to newborn AGMs. Following vaccination, we find robust levels of HA stem-specific

Abs. These HA stem-specific Abs are capable of broad recognition of group 1 HA subtypes. Overall, H1ssF + AddaVax results in reduced viral load in the lung compared to H1ssF alone. In spite of this, viral clearance following challenge is variable among animals. Evaluation of Ab quality shows the quantity of nAbs and Ab-dependent cellular phagocytosis (ADCP) effector function are associated with reduced viral load in the lung. Our results reveal the functional attributes of influenza HA stem vaccine-induced Abs that are associated with viral control in newborns. The insights garnered from these studies have direct implications for the development of influenza vaccines that can provide protection for this vulnerable population.

## Results

### H1ssF + AddaVax generates a protective response in adult AGMs

While the primary goal in the current study was to determine whether a universal HA stem vaccine is effective in newborns, we first sought to establish the utility of this vaccine in adult AGMs, as previous studies with H1ssF were performed in adult macaques[24]. This information was essential to appropriately interpret findings from newborn animals. H1ssF was delivered with the MF59-like adjuvant AddaVax, as previous studies have demonstrated that adjuvant is required for the generation of robust HA stem responses to H1ssF in IAV naïve animal models[22,24]. Adult AGMs aged 5–7 years old (equivalent to ~20–28 year old humans, Supplementary Table S1) were vaccinated intramuscularly (i.m.) with H1ssF + AddaVax or with PBS (the control group (Ctrl)) (Fig. 1A). HA stem-specific Abs were measured by ELISA using a stabilised trimeric A/New Caledonia/20/1999 HA stem protein[31]. H1ssF + AddaVax vaccinated adults had a significant increase in HA stem-specific IgM (Fig. 1B) and IgG (Fig. 1C) compared to Ctrl animals following prime (p.v.) and boost (p.b.). Modest but detectable levels of IgA were also generated (Fig. S1A). Interestingly, we observed significant generation of stem-specific IgG at d10 p.v. consistent with an early class switched plasmablast response. These data show that prime/boost vaccination with H1ssF + AddaVax in adult AGMs elicits high levels of circulating HA stem-specific Abs.

Previous studies with this vaccine have shown broad Ab reactivity across group 1 HA subtypes in adult animal models[22,24]. Thus, we evaluated the breadth of HA recognition in our vaccinated adult AGMs at the latest time point prior to challenge (d45 p.b.) (Fig. 1D). FI6V3, a human HA stem-specific monoclonal Ab (mAb) that binds the central stem epitope on group 1 and group 2 HA subtypes[32], was included as a positive control. Animals administered H1ssF + AddaVax displayed reactivity to multiple group 1 HA subtypes, with the highest recognition to H1 and H5 (Fig. 1D). Thus, the H1ssF + AddaVax prime/boost vaccination resulted in Abs that can react across group 1 HA subtypes in adult AGMs.

To determine the fine specificity of HA stem-specific Abs, we utilised a mutant NC99 HA molecule wherein an N-linked glycosylation site was introduced (Ile545Asn and a Gly547Thr (H3 numbering)) that blocks Abs binding to the central epitope, hereafter referred to as H1-N45[33]. We employed two additional NC99 HA mutants to better characterise epitope specificities, H1-N27 (Q27N and N29T) has a glycan in the anchor epitope that blocks recognition, and H1-N27/45 incorporates glycans in both the central and the anchor epitopes[26]. We used this set of mutant HA antigens to profile stem-specific Abs in the vaccinated adult AGMs. To accurately determine Abs that bind either the central or anchor epitope only, we subtracted the area under the curve (AUC) obtained with the H1-N27/45 (non-central, non-anchor) from either the H1-N27 (central binding Abs) or the H1-N45 (anchor binding Abs) (Fig. 1E and Supplementary Fig. S2). We found readily detectable Abs recognising the central region of HA stem, with one animal making a much higher response (Fig. 1E). There was limited recognition of the anchor epitope. All animals made Abs that bound epitopes outside of the central or anchor regions (non-central/anchor epitopes) (Supplementary Fig. S2).

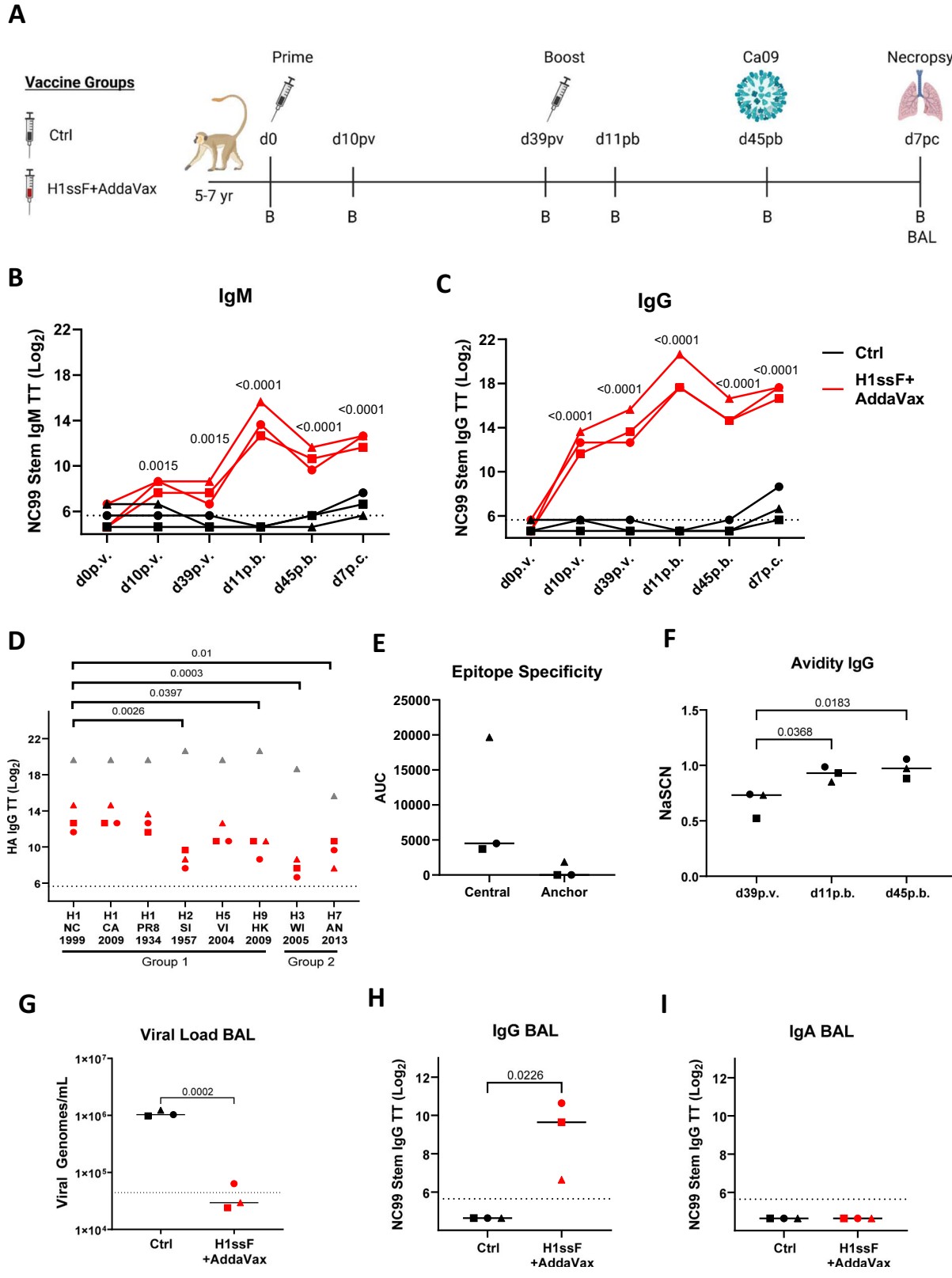

Based on a murine study that reported HA stem-specific Abs can increase the susceptibility to autoimmune diseases[34], we tested whether H1ssF+AddaVax elicited increases in autoreactive Abs through the measurement of IgM and IgG Abs specific to autologous nuclear antigens (ANA). There was no indication of vaccine-mediated induction of ANA Abs (Supplementary Fig. S3).

We next measured the avidity of HA stem-specific IgG over time in adults that received H1ssF + AddaVax. Increased avidity is associated with GC formation and somatic hypermutation[35]. Boosting with H1ssF + AddaVax resulted in a rapid rise in HA stem-specific Ab avidity at d11 p.b.; this did not increase further at d45 p.b. (Fig. 1F).

**Fig. 1 | Vaccination with H1ssF+AddaVax elicits robust HA stem-specific Abs with broad HA recognition that are associated with viral clearance in adult AGMs after Ca09 challenge.** Adult AGM vaccination and sampling schedule (**A**). Circulating levels of NC99 stem-specific IgM (**B**) and IgG (**C**) were assessed by ELISA. HA-specific IgG cross-reactivity was measured against group 1 and group 2 HA molecules in the plasma at d45 p.b. The stem-reactive mAb FI6V3 (grey triangle) served as a positive control (**D**). Circulating levels of IgG binding to the full-length NC99 HA, H1-N27, H1-N45, or H1-N27/45 were assessed by ELISA at d45 p.b. Central and anchor Abs were quantified by subtracting AUC for H1-27/45 from the H1-27 (central) or H1-N45 (anchor) AUC values (**E**). NC99 stem-specific IgG avidity was measured by determining the NaSCN concentration that gave a 50% reduction in optical absorbance compared to the untreated sample (**F**). Adult AGMs were challenged with $5 \times 10^7$ TCID$_{50}$ of Ca09 H1N1 at d45 p.b. Viral genomes were measured in BAL collected at d7 p.c. (**G**). NC99 HA stem-specific IgG (**H**) and IgA (**I**) were measured in the BAL at d7 p.c. The dotted line represents the limit of detection (LOD). Threshold titre (TT) is defined as the highest dilution resulting in an OD$_{450}$ greater than 3X the assay background. Ctrl ($n = 3$), H1ssF + AddaVax ($n = 3$). Symbols represent individual animals. The line in each column represents the median. Statistical significance: two-way ANOVA with Tukey's post hoc analysis (**B, C, H**), a one-way ANOVA with either Tukey's (**D, F**) post hoc analysis, or a paired (**E**), or unpaired two-tailed $t$ test (**G, I, J**). Not significant $p \geq 0.05$ (not indicated on graph). Figure 1A was created in BioRender. Alexander-Miller, M. (2025) https://BioRender.com/017wbod. Source data are provided as a Source Data file.

To assess the ability of this vaccine to provide protection in the adult AGM model, we challenged animals with the clinically relevant H1N1 pandemic strain A/California/07/2009 (Ca09) ($5 \times 10^7$ TCID$_{50}$). We assessed HA stem-specific Ab and viral load in the lungs using bronchoalveolar lavage (BAL) samples collected at d7 p.c. Virus was significantly reduced in the lungs of H1ssF + AddaVax animals at d7 p.c., with 2 of 3 animals having cleared virus (Fig. 1G). We observed higher levels of HA stem-specific IgG in the BAL of H1ssF+AddaVax adults compared to Ctrl adults at d7p.c. (Fig. 1H). No HA stem-specific IgA was detected in the BAL (Fig. 1I). Thus, H1ssF + AddaVax vaccine elicited robust stem-specific, broadly reactive Ab responses and promoted enhanced viral clearance in the lungs of adult AGMs. These data establish the ability of this vaccine approach to generate protective responses in the AGM model.

HA stem-specific Abs have been shown to promote clearance through both neutralising activity[36,37] and Fc-mediated effector function[38,39]. In adult naive animal models (3 doses)[22,24] and in human adults (2 doses)[25], the H1ssF vaccine elicits Abs that have both neutralising and Ab-dependent cellular cytotoxicity (ADCC) activity[25].

We evaluated the functional attributes, i.e., neutralising capacity, ADCC, Ab-dependent complement deposition (ADCD), and ADCP of the HA stem-specific Abs generated in adult AGMs following H1ssF +AddaVax prime/boost vaccination and after challenge with Ca09 H1N1. Neutralising capacity was assessed using a set of replication-restricted reporter influenza viruses (H1N1 A/New Caledonia/20/1999, H1N1 A/California/07/2009, and H5N1 A/Vietnam/1203/2004)[40]. Interestingly, nAbs were not detected at d45 p.b. (Fig. 2A). However, neutralisation titres against the vaccine-matched NC99 were readily detected at d7 p.c. in animals administered H1ssF + AddaVax, but not in Ctrl animals (Fig. 2A). These data indicate a memory response was generated with vaccination that resulted in the rapid production of nAb after viral challenge. Neutralising Abs to the challenge virus (Ca09) were also detected, albeit at lower levels than to NC99 (Fig. 2B). Remarkably, all of the vaccinated adults also had nAb to H5N1 following challenge (Fig. 2C).

Next, we explored the ability of vaccine-elicited Abs to facilitate Fc-mediated effector function through ADCC, ADCD, and ADCP. Surprisingly, no ADCC, ADCD, or ADCP activity was detected in plasma from H1ssF + AddaVax vaccinated adults at d45 p.b. (Fig. 2D–F). However, there was a significant increase in all three Fc-mediated functions at d7 p.c. in vaccinated adults. Overall, these data support a rapid recall response in vaccinated adult animals that is associated with clearance and exhibits a broad array of Ab effector functions.

## H1ssF + AddaVax promotes significant levels of HA stem-specific Abs in newborn AGMs that are broadly reactive

Having demonstrated the utility of H1ssF +AddaVax in adults, we sought to determine the capacity for newborns to respond to this vaccine. The AddaVax adjuvant used here is similar to the licensed MF59, which has a demonstrated safety profile in human newborns in the context of an HIV vaccine delivered at birth, 2w, 8w, and 20w[41]. AGMs aged 3–5 days old (equivalent to ~12–20 day old humans,

Supplementary Table S2) received an H1ssF prime/boost regimen with or without AddaVax (Fig. 3A). As these studies are the first to explore the response to H1ssF in newborn NHPs, we selected a dose (40 μg) previously administered to adult ferrets[22], which are approximately the same weight as our newborn AGMs. As a non-vaccinated control, newborns received PBS or an mRNA lipid nanoparticle vaccine expressing luciferase (leveraged from another ongoing study). These animals are referred to as the control group (Ctrl) moving forward. All animals were challenged with H1N1 Ca09 ($1 \times 10^7$ TCID$_{50}$) at d41/45 p.b. to assess the ability of this vaccine to promote responses that can reduce viral load.

Infants were monitored for 15 minutes following vaccination; no sign of adverse reaction was noted during this period. At 24 h p.v., infants were assessed for changes at the vaccination site as well as general health characteristics (e.g., alert, latching on to mothers). No vaccine-induced changes were observed. As a measure of systemic inflammation, we assessed C-reactive protein (CRP) on d1 p.v. A CRP level lower than 10 mg/L is considered low, with levels greater than 10 mg/L indicating an overly robust inflammatory reaction[42]. All animals were below 10 mg/L (median = 0.82 mg/L, Supplementary Fig. S4A). There was no statistically significant difference in the CRP levels between the vaccine groups. We also measured rectal temperatures on d1 p.v. All were within the normal range (Supplementary Fig. S4B). There was also no difference in weight gain across the groups over the course of the experiment (Supplementary Fig. S4C). These data suggest that the vaccine does not induce adverse systemic reactions.

Newborns receiving H1ssF + AddaVax exhibited a significant increase in the quantity of HA stem-specific IgM (Fig. 3B) and IgG (Fig. 3C) at all-time points following vaccination compared to Ctrl animals. Newborns administered H1ssF in the absence of adjuvant were not statistically significant from Ctrl newborns. Interestingly, in 4 of 8 H1ssF + AddaVax vaccinated newborns, we found detectable stem-specific IgG responses as early as d10 p.v. Interestingly, as was observed in adults, we found increased levels of stem-specific IgA in the plasma in the H1ssF + AddaVax vaccinated animals compared to Ctrl and H1ssF vaccinated animals (Supplementary Fig. S1B).

Next, we determined the capacity of HA stem-specific Abs to recognise divergent subtypes of HA, assessing the time point at which the response was most mature in our experimental regimen, d41/45 p.b. (Fig. 3D). The H1ssF + AddaVax elicited Abs with the capacity for broad recognition against group 1 HAs (H1, H2, H5, and H9) when compared to Ctrl and H1ssF vaccinated newborns. Recognition of Ca09 H1 HA was strongest, with a titre similar to that for NC99, the stem used in the vaccine. Recognition of H1 from PR8, H5, and H9 (group 1) was robust, although reduced compared to NC99. The weakest recognition among the group 1 HAs was H2. Recognition of group 2 HA by Abs from the H1ssF+AddaVax newborns was not above Ctrl animals (Fig. 3D). Although not statistically significant as a group, there were select H1ssF + AddaVax animals that exhibited higher levels of H7 binding, suggesting their repertoire may allow recognition of this group 2 HA.

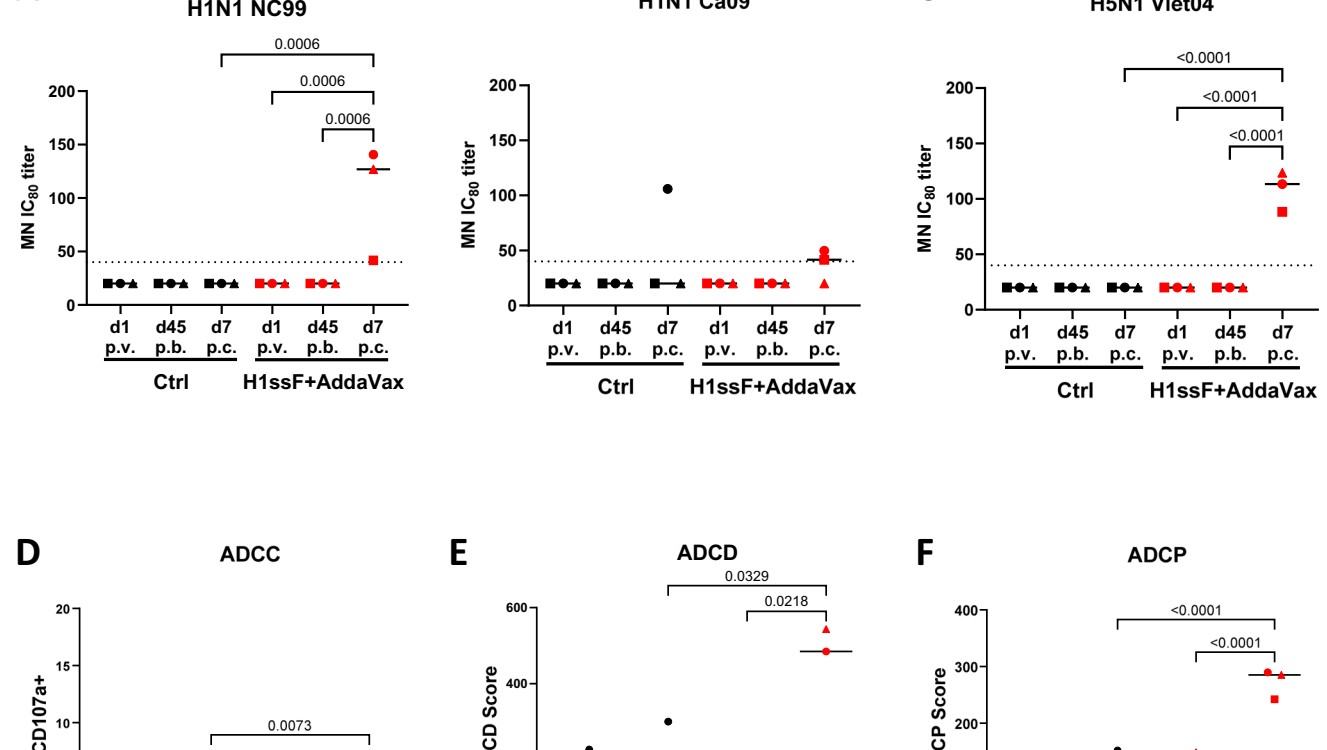

**Fig. 2 | Antibody quality increases after challenge in adult AGMs that were vaccinated and boosted with H1ssF + AddaVax.** Neutralising $IC_{80}$ titres against H1N1 A/New Caledonia/20/1999 (**A**), H1N1 A/California/07/2009 (**B**), and H5N1 A/Vietnam/1203/04 (**C**) were assessed by reporter-based microneutralization (MN) assays. The MN $IC_{80}$ titre was measured at the plasma dilution at which 80% of virus infection was inhibited. ADCC activity against HA Ca09 was measured using KHYG-1 cells expressing rhesus macaque CD16. ADCC activity was measured as the percentage of CD107a+ KHYG-1 cells present after sequential gating of cells by FSC/SSC and Zombie Violet negativity (**D**). ADCD scores were determined by calculating the percentage of HA FluoSpheres beads that were C3 + × GMFI/1000 at the highest dilution (1:5) (**E**). ADCP was measured by THP-1 uptake of Ab-opsonised HA Fluo-Spheres beads. THP-1 were gated through FSC/SSC and singlet gates prior to FITC bead analysis. Phagocytosis scores were determined by measuring the percentage of bead-positive THP-1 cells×the the geometric mean fluorescence/1000 at the 1:800 dilution. (**F**). The dotted line represents the limit of detection (LOD) for each assay. Ctrl ($n = 3$), H1ssF + AddaVax ($n = 3$). The line in each column represents the median. Statistical significance was determined using a one-way ANOVA with a Fisher's LSD post hoc analysis. Not significant $p = > 0.05$ (not indicated on graph). Source data are provided as a Source Data file.

Given the outbred nature of the newborn AGMs, we were curious whether the binding pattern across the HA molecules differed among individual animals. To address this, we calculated the efficiency of recognition of each HA type vs. NC99 (threshold titre of HA X:threshold titre of HA NC99). This ratio allowed us to normalise for any differences in total HA stem-specific Abs among animals. We observed diverse binding patterns (Fig. 3E), with some infants binding H5 with high efficiency (e.g., 2661 (○)), while others more efficiently bound H9 (e.g., 2698 (□)). Together, these data show that the vaccine-elicited stem-specific Abs bind diverse group 1 HA subtypes; however, the efficiency of binding across the HA molecules in relation to the parent strain differed among animals. The divergent binding patterns suggest there may be differences among newborns in the quality of Abs generated, i.e., avidity, or differences in epitope specificity represented within the polyclonal response.

To test the latter, we used the glycan mutant HA molecules described above to quantify the relative response to the central and anchor epitopes at d41/45 p.b. Vaccinated newborns generated a strong Ab response to the central epitope compared to anchor or non-central/anchor epitopes (Fig. 3F, primary data Supplementary Fig. S5). We noted that the quantity of central stem Ab varied considerably

among the H1ssF + AddaVax animals (Fig. 3F). In contrast to adult animals, the response to regions outside of the central or anchor was modest.

One mechanism that has been postulated to contribute to the subdominance of the stem response is steric hindrance in accessing this region of HA on the surface of the virion[21]. To understand the capacity of the Abs generated in response to vaccination with the stem region to bind HA on an intact virus, we assessed recognition of the purified PR8 virus (Supplementary Fig. S6A). Recognition of the PR8 virus correlated with that of PR8 HA (r = 0.86, $r^2 = 0.74$, $p = 0.006$) (Supplementary Fig. S6B). This finding supports the ability of Abs elicited by H1ssF to recognise HA on the influenza virion.

We also assessed the capacity of stem-specific Abs elicited by H1ssF to recognise drifted H1 variants using the 2023-2024 Fluzone quadrivalent vaccine containing A/Victoria/4897/2022 (H1N1), A/Darwin/9/2021 (H3N2), B/Phuket/3073/2013 (B Yamagata lineage), and B/Michigan/01/2021 (B Victoria lineage). We found significant binding to the Fluzone vaccine in plasma from H1ssF + AddaVax vaccinated newborns compared to the H1ssF and Ctrl animals (Supplementary Fig. S6C). A strong correlation was observed between Fluzone-specific IgG and NC99 stem-specific IgG in H1ssF + AddaVax newborns

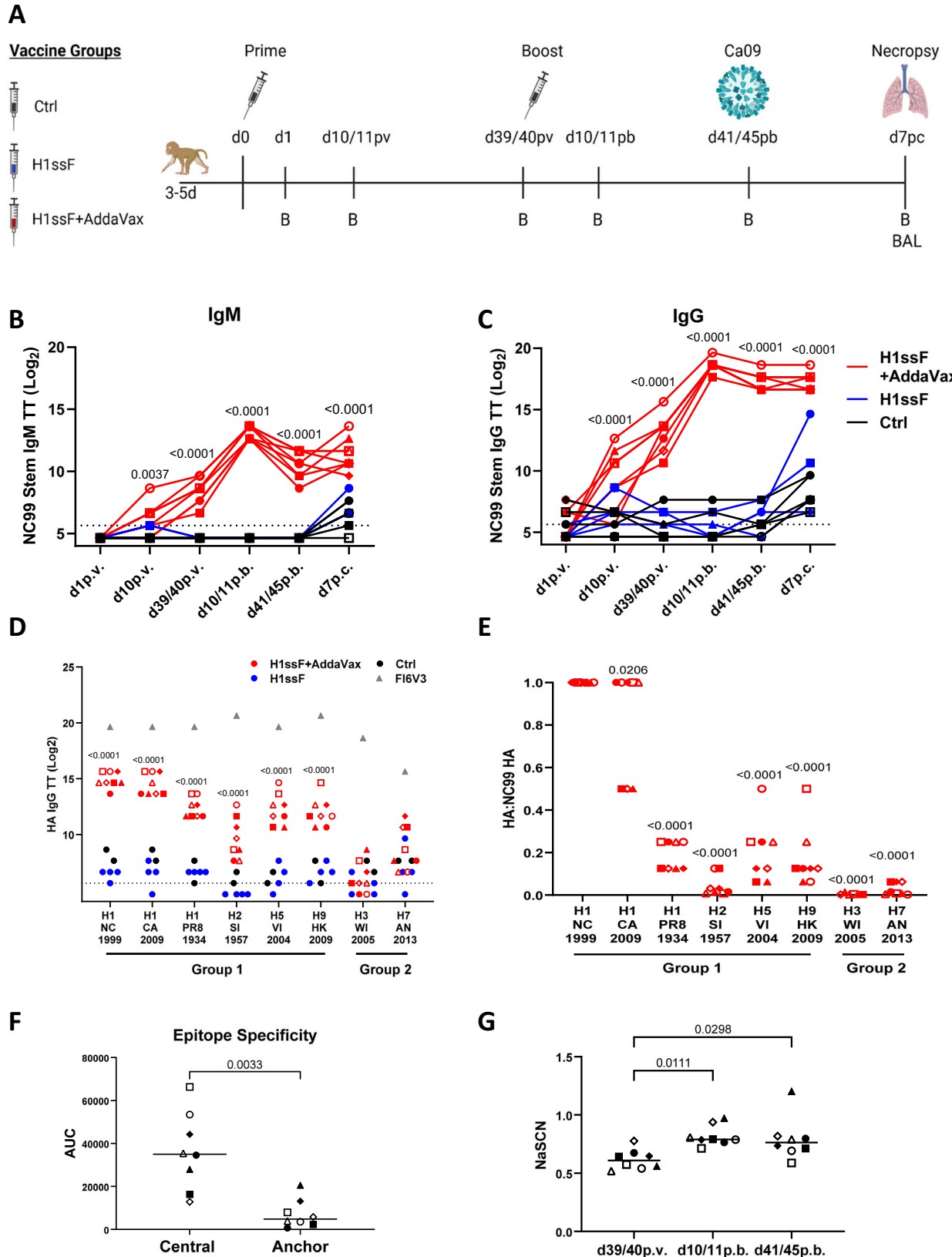

(Supplementary Fig. S6D). These data show the vaccine-elicited stem-specific IgG is capable of recognising modern HA molecules. We would predict this is via binding to H1 in the vaccine, based on our finding of limited group 2 recognition by these Abs. As in adults, there was no indication of a vaccine-mediated induction of ANA-specific IgM or IgG in newborn AGM (Supplementary Fig. S7).

Finally, we measured the avidity of HA stem-specific IgG over time in newborns that received H1ssF+AddaVax (Fig. 3G). Boosting resulted in increased avidity, with a rapid rise in HA stem-specific Ab avidity by d10/11 p.b. Overall, these data demonstrate vaccination of newborn AGMs with H1ssF + AddaVax results in uniformly high levels of HA stem Ab. These Abs are broadly

**Fig. 3 | Vaccination with H1ssF+AddaVax promotes cross-reactive HA stem-specific Abs in newborn AGMs.** Newborn AGM vaccination and sampling schedule (**A**). Circulating levels of NC99 stem-specific IgM (**B**) and IgG (**C**) were assessed by ELISA. IgG reactivity was measured against group 1 and group 2 HA molecules in the plasma at d41/45 p.b. The stem-reactive mAb FI6V3 was included as a positive control (**D**). The ratio of the TT for each HA divided by that for NC99 HA was calculated as a measure of the efficiency of recognition of each HA molecule (**E**). Circulating levels of IgG that binds to full-length NC99 HA, H1-N27, H1-N45, or H1-N27/45 were assessed by ELISA at d41/45 p.b. The quantity of central and anchor region Abs was determined by subtracting the AUC for H1-27/45 from that of H1-27 (central) and H1-N45 (anchor) (**F**). NC99 stem-specific IgG avidity was measured by determining the concentration of NaSCN that resulted in a 50% reduction in optical absorbance compared to the untreated sample (**G**). The dotted line represents the LOD. TT was defined as the highest dilution resulting in an $OD_{450}$ greater than 3X assay background. Ctrl ($n = 5$), H1ssF ($n = 4$), H1ssF + AddaVax ($n = 8$). Symbols represent individual animals and are maintained throughout the datasets. The line in each column represents the median (**F**, **G**). Statistical analysis: two-way ANOVA with Tukey's post hoc analysis (**B**–**D**, **G**), a one-way ANOVA with a Dunnet's (H1 NC99 is the control) (**E**), or a two-sided paired t test (**F**). Statistics represent H1ssF + AddaVax vs. Ctrl. (Fig. 3B, C, and D). Ctrl vs. H1ssF was not significant. Statistical analysis in Fig. 3E is for each HA vs. WT NC99 HA. Not significant $p = >0.05$ (not included on graph). Figure 3A was created in BioRender. Alexander-Miller, M. (2025) https://BioRender.com/vqpx526. Source data are provided as a Source Data file.

## Vaccination with H1ssF + AddaVax results in reduced virus following Ca09 challenge in a subset of newborns

To evaluate the capacity for H1ssF + AddaVax to control infection in the newborn AGMs, we challenged animals with H1N1 Ca09 ($1 \times 10^7$ TCID$_{50}$) at d41/45 p.b. Viral load was measured by RT-PCR in the lung at d7 p.c. A significantly lower level of virus was found in the lungs of animals vaccinated with H1ssF + AddaVax compared to H1ssF alone, with two animals exhibiting markedly superior clearance, 2661 (○) and 2698 (□) (Fig. 4A). No significant difference was observed between Ctrl and H1ssF animals.

A strong inflammatory environment in the lung following influenza virus infection is associated with increased disease[43]. In children with influenza, a higher level of circulating IL-6 is correlated with greater disease severity[44]. In our study, newborns that received H1ssF +AddaVax had significantly lower IL-6 compared to both H1ssF and Ctrl animals (Fig. 4B). Overall, these data show that H1ssF+AddaVax vaccination resulted in lower viral load and lower levels of IL-6 in newborn AGM. Further, they suggest that, although the vaccine induced high levels of Ab in all newborns, there were differences in the ability of the vaccine response to promote viral clearance in the lung.

## The level of HA stem-specific Ab in the lung does not predict viral load in newborn AGMs

One possibility to explain the differences in clearance was the level of stem-specific IgG present in the respiratory tract. Although vaccination resulted in higher levels of stem-specific IgG in the BAL, this was not the determinant of viral clearance, as not all animals that had higher levels of Ab had lower viral loads (Fig. 4C). IgA was not detectable in the BAL at this time (Fig. 4D). While it is possible that IgA is present at a level that is below the limit of detection in our assay, it seems unlikely that such a level would be sufficient for the clearance observed in the subset of newborns.

## Reduced viral load in the lower respiratory tract of newborn AGMs is correlated with Ab function

Given the lack of correlation between Ab quantity and viral clearance in the newborns, we evaluated the functional attributes of the HA stem-specific Abs generated in response to H1ssF + AddaVax. This has not, to our knowledge, been explored in newborns. Neutralising Abs against the vaccine matched NC99 virus were present at d41/45 p.b in 5 of 8 newborns that received H1ssF + AddaVax (Fig. 5A). Three newborns had detectable titres to Ca09 (Fig. 5B) and 2 to H5N1 (Fig. 5C). Importantly, the 2 newborns with the highest neutralising titres across all three viruses (2661 (○) and 2698 (□)) also had the lowest viral titres in the BAL at d7 p.c. (Fig. 4A).

Neutralising Abs against all three strains increased following challenge in H1ssF + AddaVax vaccinated infants (Fig. 5A–C). Although there may be some HA head directed nAbs in the post challenge response, we expect the majority of the nAb in the H1ssF + AddaVax animals is a stem induced vaccine response, as 7 out of 8 animals are capable of neutralising heterosubtypic H5N1 (Fig. 5C). Interestingly, when we compared the level of central binding IgG Abs at d41/45 p.b. against NC99 nAb d41/45 p.b. (Fig. 5D), we observed a positive correlation ($r = 0.94$, $p = 0.0012$), consistent with a model where higher levels of central binding stem specific Abs contribute to higher nAb titres. We performed another correlation analysis plotting nAb to NC99 present at d7 p.c. against viral load in the BAL for all newborns, finding higher nAb against NC99 was also associated with lower viral load ($r = -0.68$, $p = 0.0035$) (Fig. 5E). These data are consistent with a role for nAbs as a component of viral clearance in newborn AGMs. Newborns vaccinated with H1ssF in the absence of adjuvant and Ctrl animals had lower nAb titres against H1 Ca09 at d7 p.c., while the presence of nAb was variable for H1 NC99 and absent for H5 Viet04 (Fig. 5A–C).

Lastly, we explored the ability of these Abs to promote ADCC, ADCD, and ADCP. Neither ADCC nor ADCD activity was detected in H1ssF+AddaVax newborns prior to challenge (d41/45 p.b.) (Fig. 6A and B). In contrast, ADCP activity was readily detected (Fig. 6C). Challenge resulted in a statistically significant increase in all three activities in newborns who had received H1ssF+AddaVax. Interestingly, ADCP was also robustly induced in naive newborns following challenge (Fig. 6C). Given the correlation between nAb and central epitope Abs, we probed the relationship between binding to the central epitope and ADCP, finding no significant correlation between these Abs and ADCP (Supplementary Fig. S8). We speculate that this is due to the ability of Ab binding to any site on the HA molecule to facilitate ADCP.

To evaluate the relationship between ADCP activity and viral load, we plotted the vaccine-induced ADCP scores at d41/45 p.b. against viral load in the BAL at d7 p.c. for each newborn. We found these two readouts were negatively correlated ($r = -0.76$, $r^2 = 0.58$, $p = 0.0006$), with higher ADCP scores associated with lower viral load (Fig. 6D). Interestingly, the newborn that had the highest ADCP score pre-challenge (2661 (○)) also had the lowest viral load in the lower respiratory tract (Fig. 4C). Overall, these data support a model wherein neutralising capacity and ADCP activity can mediate influenza viral clearance in newborn AGMs.

## Discussion

The development of a broadly protective universal vaccine would represent a major step forward in combating infections arising from both seasonal and pandemic influenza outbreaks. With promising outcomes emerging from clinical trials conducted in adults, it is essential that we assess how the immune system of newborns and young infants responds to an HA stem-based vaccine. In addition, understanding the properties of Abs elicited by such a vaccine that can mediate protection in the context of the newborn is crucial. Here, we evaluated the effectiveness of an HA stem nanoparticle vaccine in newborn NHPs, the most relevant preclinical model for testing vaccine responses in this age group.

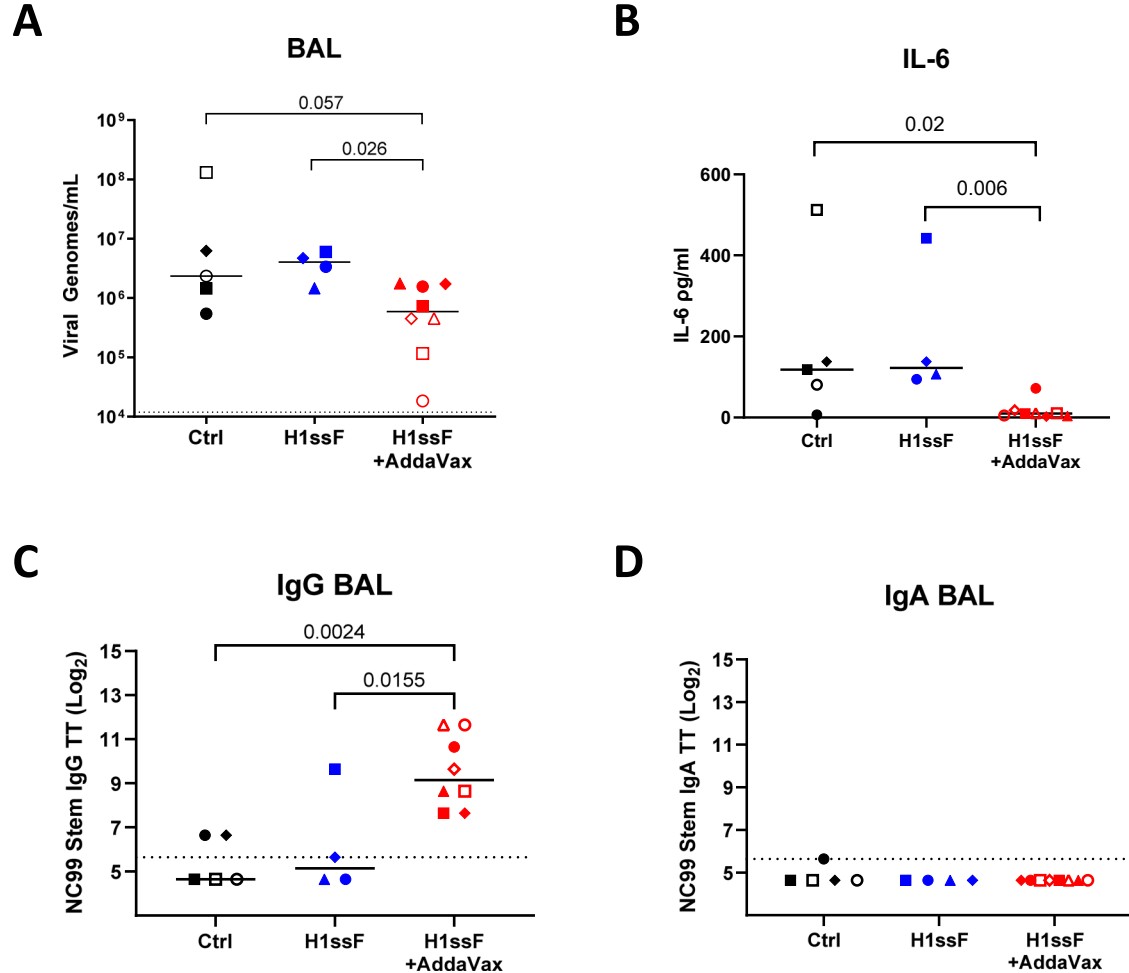

**Fig. 4 | Vaccination with H1ssF+AddaVax resulted in viral clearance following H1N1 challenge in a subset of newborn AGMs.** Newborn AGMs were challenged with $1 \times 10^7$ TCID$_{50}$ of Ca09 H1N1 at d41/45 p.b. Viral genomes were measured in the BAL at d7p.c. (**A**). IL-6 levels were measured in the BAL at d7 p.c. (**B**). NC99 stem-specific IgG (**C**) and IgA (**D**) were assessed in the BAL at d7 p.c. in newborn AGM. The dotted line represents the LOD for the assay. The line in each column represents the median. Statistical significance was determined by using a one-way ANOVA with pairwise comparisons between groups (**A**), a non-parametric Wilcoxon rank scores with Kruskal-Wallis analysis (**B**), or a one-way ANOVA with Tukey's post hoc analysis (**C**, **D**). Not significant $p \geq 0.05$ (not indicated on graph). Source data are provided as a Source Data file.

To ensure the utility of this vaccine in our model, we first immunised adult AGMs with H1ssF + AddaVax, as previous assessments of this vaccine were performed in adult cynomolgus macaques. Similar to other adult models, H1ssF + AddaVax resulted in the generation of broadly reactive HA stem-specific Abs. We found this response was associated with reduced viral load in the lung after challenge, a readout not previously assessed in NHPs. These data established the ability of this vaccine to promote protective responses, providing confidence that results obtained in the newborn could be attributed to age and not the AGM model employed.

Vaccination of newborns with H1ssF + AddaVax resulted in robust stem-specific responses. Abs elicited by H1ssF + AddaVax exhibited strong binding against group 1 HAs. When we investigated the epitope specificity (central vs. anchor region) of the elicited Abs, we discovered the vast majority of the response was directed to the central epitope. Unexpectedly, although all newborns administered H1ssF + AddaVax generated a high level of Ab, they displayed heterogeneity in their capacity to reduce virus in the lung. Interrogation of the capabilities of the elicited Abs revealed strong correlations between both nAbs and ADCP activity with reduced viral load in the lung.

In our analysis of nAb production following administration of H1ssF + AddaVax, we found that only a subset of newborns had Abs

capable of this function. One explanation for this finding could be differences among newborn GC responses. GC formation requires the spatial and temporal coordination between several subsets of immune cells, which has been shown in multiple studies to be hampered in infants (for review see ref. [10]). While the GC response has not been assessed following administration of H1ssF to adult NHPs, unexpectedly, studies utilising a similar ferritin nanoparticle vaccine platform containing full length HA administered with AddaVax to adult pigtail macaques found no evidence of improved GC B cells or Tfh cell responses in the vaccine draining lymph node[45]. Thus, the regulation of GC formation by this vaccine platform requires further investigation. In humans, H1ssF delivered in the absence of adjuvant resulted in robust plasmablast responses, improved sustained memory B cell responses, and production of nAbs[26]. However, it seems likely this reflects the expansion of memory cells present in humans prior to vaccination.

Encouragingly, we observed rapid nAb development after challenge in all newborns that received H1ssF + AddaVax. Whether this is the result of a third antigen exposure or environmental signals associated with infection is unclear. These data suggest that despite the lack of detectable nAb responses after prime/boost in some newborns, H1ssF + AddaVax promoted a stem-specific memory response in all

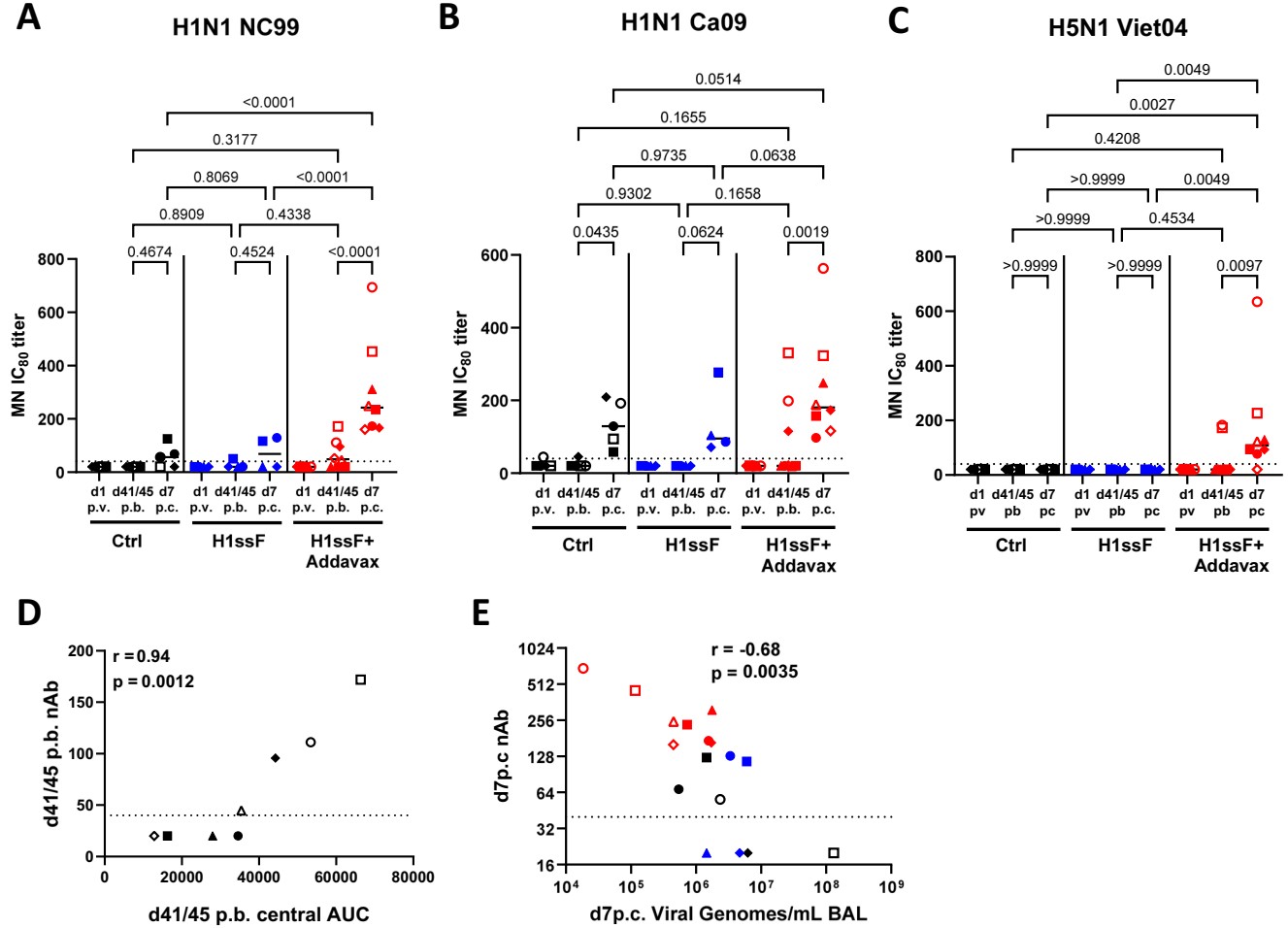

**Fig. 5 | Vaccination with H1ssF+AddaVax results in broadly neutralising antibody in a subset of newborn AGM that is correlated with viral clearance.** Neutralising Ab was measured using a reporter-based microneutralization (MN) assay at d1 p.v., d41/45 p.b., and d7 p.c. Neutralising $IC_{80}$ titres against H1N1 A/New Caledonia/20/1999 (**A**), H1N1 A/California/07/2009 (**B**), and H5N1 A/Vietnam/1203/ 04 (**C**) were quantified. A Spearman's correlation analysis was performed on A/New Caledonia/20/1999 nAb $IC_{80}$ titre from d41/45 p.b and the NC99 central epitope IgG AUC from d41/45 p.b in H1ssF + AddaVax newborns (**D**). A Spearman's correlation analysis was performed on A/New Caledonia/20/1999 nAb $IC_{80}$ titre and viral genomes/mL in the BAL at d7 p.c. (**E**). Ctrl (n = 5), H1ssF (n = 4), H1ssF+AddaVax (n = 8). The dotted line represents the limit of detection (LOD) for the assay. The line in each column represents the median. Statistical significance was determined using a one-way ANOVA with a Fisher's LSD post hoc analysis. Not significant $p \geq 0.05$ (not indicated on graph). Source data are provided as a Source Data file.

newborns that was capable of rapid recall and nAb production after challenge. Rapid recall responses, as assessed by either quantity or subclass changes, have been observed in other models, e.g., mouse studies using a split influenza A (H3N2) vaccine[46] or administration of a COBRA-based influenza vaccine[47]. In adult humans, there is also evidence of a rapid recall response following H1ssF vaccination, as plasmablasts peak in circulation between d5 and d7 pv[26].

The Abs produced in newborns following vaccination with the NC99 stem were capable of neutralising multiple strains, including an H5N1 virus. In adult humans, H1ssF vaccination resulted in Abs to both the central and membrane proximal anchor epitopes, with the majority of these Abs binding the central epitope[26]. Follow-up analysis found Abs that bound the central epitope were more broadly neutralising across group 1 strains than those binding the anchor epitope, with particularly strong neutralising potency observed against H5[26]. In our studies, after prime/boost vaccination, newborns that had the highest nAb to H5 also had a higher quantity of central stem-specific Abs. Surprisingly perhaps, newborn animals appeared to generate Abs to the central epitope with higher efficiency than adults (7.8 fold higher level).

In addition to nAb, H1ssF + AddaVax promoted Abs with ADCP activity in newborns. The presence of ADCP competent Abs was associated with improved viral clearance. In contrast, neither ADCC nor ADCD activity was detected following prime/boost vaccination. The correlation we observed between ADCP activity and decreased influenza virus load in vaccinated newborns is in agreement with previous studies showing Abs that exert ADCP effector function can play an important role in the clearance of influenza virus[48–51]. We observed boosted ADCP as well as measurable ADCC and ADCD activity in H1ssF + AddaVax newborns after challenge, suggesting infection and/or a third antigen exposure drives production of a multifunctional Ab pool from memory cells present in newborns. The development of effector activity within the Ab pool generated following vaccination of newborns is likely dependent on both vaccine and adjuvant. For example, in a study from De Paris and colleagues, ADCC activity varied with adjuvant following vaccination and boosting with Env gp120 monomers in newborn rhesus macaques[52]. In adults, individuals who received the H5N1 vaccine with MF59 exhibited increases in Abs capable of promoting virus uptake by neutrophils and complement-activating activity compared to individuals who received alum-adjuvanted or non-adjuvanted vaccine[53]. Interestingly, Abs from H5N1 + MF59 vaccinated individuals had limited ADCC effector function[53,54]. A fuller understanding of the ability of adjuvants to regulate these properties in the responding Ab pool may provide insights

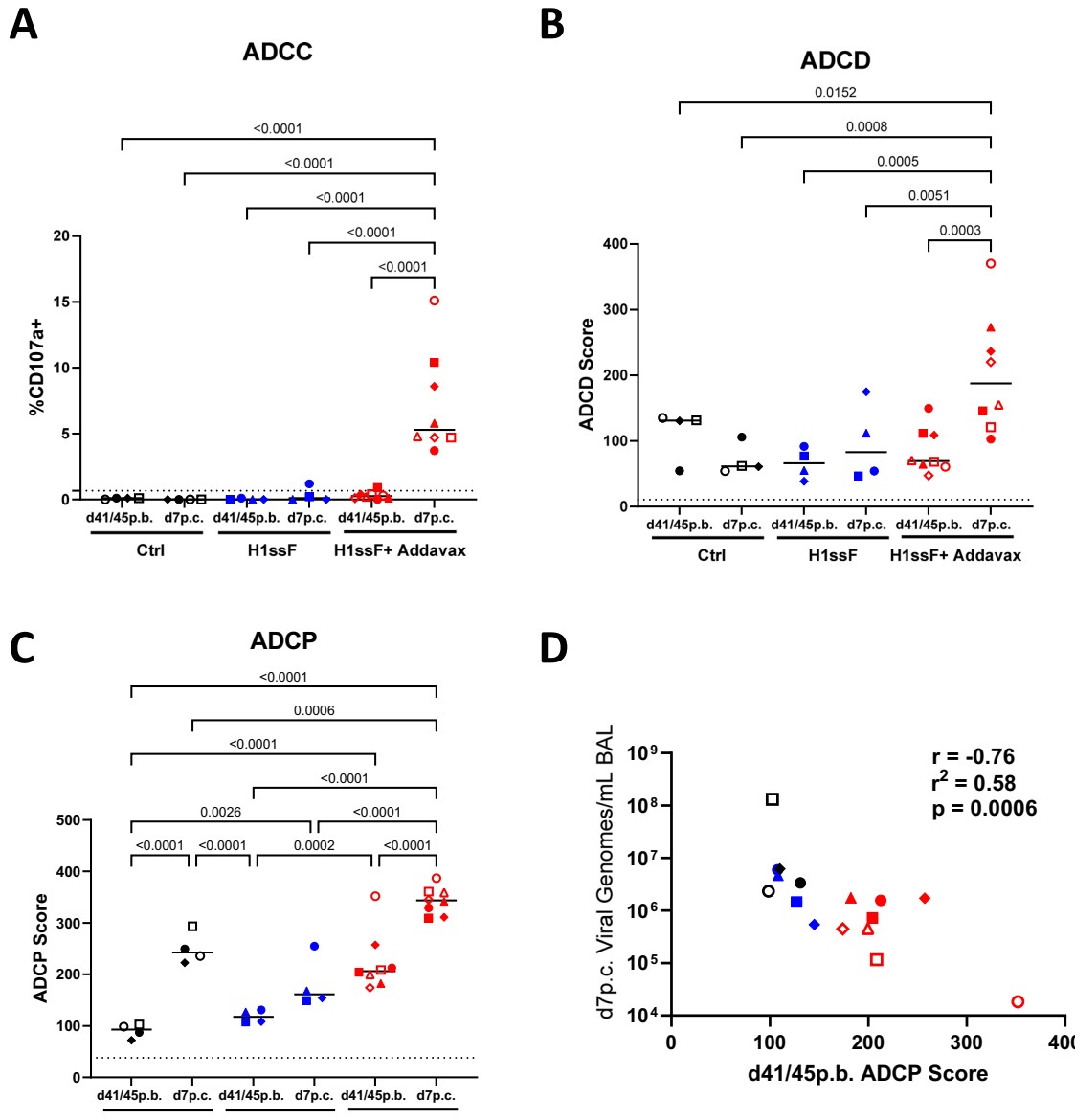

**Fig. 6 | Vaccination of newborn AGM with H1ssF+AddaVax results in antibody with ADCP activity that is correlated with viral clearance.** Plasma samples were assessed for Ab function at d41/45 p.b. and d7 p.c. in newborn AGM. ADCC activity was quantified using the percent of CD107a + KHYG-1 cells. Cells were sequentially gated as follows: FSC/SSC and viability staining, followed by CD107a positivity (**A**). ADCD scores were determined as follows: percent C3 + beads × GMFI/1000 at a 1:5 dilution of plasma (**B**). ADCP scores were determined using the percent of bead-positive THP-1 cells X the geometric mean/1000 at a 1:800 dilution of plasma. THP-1 were gated through FSC/SSC and singlet gates prior to FITC bead+ analysis (**C**). A Pearson correlation analysis was performed on viral genomes/mL in the BAL at d7 p.c. versus d41/45 p.b. ADCP scores (**D**). Ctrl ($n = 4$), H1ssF ($n = 4$), H1ssF + AddaVax ($n = 8$)). The dotted line represents the limit of detection (LOD) for the assay. The line in each column represents the median. Statistical significance was determined using a one-way ANOVA with a Fisher's LSD post hoc analysis. Not significant $p \geq 0.05$ (not indicated on graph). Source data are provided as a Source Data file.

into the design of vaccine approaches capable of promoting Abs with the desired range of Fc effector activities.

Based on the differences in Ab function that we observed in the adults and newborns following vaccination, we directly compared the quantity and quality of the Ab response in these two groups. In general, stem-specific IgG was relatively similar over the time course assessed. However, we did note higher levels at d10 p.v. in the adults (Supplementary Fig. S9A), consistent with a more rapid response to vaccination. Surprisingly, newborns had higher levels of IgA in the plasma compared to adults (Supplementary Fig. S1), with a strong positive correlation between the level of stem-specific IgG and IgA (Supplementary Fig. S1) ($r = 0.91$, $r^2 = 0.83$, $p = <0.0001$). These data add to the

growing number of examples of adjuvanted protein based vaccines, e.g., hepatitis B surface antigen (which is approved for human newborns), HIV gp120, SARS-CoV-2 spike, that can elicit robust Ab responses in newborns and young infants[52,55–57]. In this regard, similar to our findings, even when a smaller dose of a SARS-CoV-2 spike protein vaccine was administered, infant rhesus macaques generated Ab that reached a magnitude similar to adults[56].

Newborns in our study had higher levels of nAb and ADCP activity compared to adults at d7 p.c. (Supplementary Fig. S1). These data demonstrate that the H1ssF + AddaVax has the capacity to elicit functional Ab responses in the newborn that are qualitatively different than adults. The ability of infants to generate a high-quality Ab response was

also observed in a recent study from Permar and colleagues, wherein infant rhesus macaques generated broadly neutralising Abs at a level similar to adults following vaccination with an HIV Env protein-based vaccine[57]. Thus, with platform and adjuvant optimisation, newborns can induce potent and protective responses following vaccination.

As noted above, ADCP activity was variable in newborns following vaccination. The Ab subclass is a potential possibility to explain differences in ADCP activity. At present, validated Abs to quantify IgG subclass in AGMs are not available. The development of such reagents would facilitate these analyses. Studies evaluating adjuvant-dependent effects on vaccine-elicited Abs have reported that Fc glycosylation determines the effector functions present[58]. Thus, differences in glycosylation of the Fc region may contribute to the Ab effector functions present.

In this study, we have shown that HA stem-specific nAb and ADCP activity appear to be important predictors of viral clearance. Given the heterogeneous response among the newborns with regard to these Ab properties, an important future goal is the development of vaccine approaches that uniformly induce Abs with these functions. One strategy may simply be an additional boost. Alternatively, increasing the dose or modulating the duration of Ag delivery may also improve GC responses and thus Ab quality in the newborn[59,60]. Finally, using alternative adjuvants or a combination of AddaVax and TLR agonists may provide further improvement to the newborn immune response. One candidate that we would expect to have beneficial effects are TLR7/8 agonists, which we and others have shown promote improvements in multiple aspects of the newborn immune response[30,55,61–68].

Identifying the optimal dose for human infants will also be a crucial step. In the first clinical trial carried out with the H1ssF vaccine, the HA content was matched to commercial influenza vaccines[69]. Historically, infants receive smaller doses of vaccines than adults. Infants (≥ 6 months) administered seasonal inactivated influenza receive half of the standard dose. We expect our newborn AGM studies to provide useful information for the design of initial trials in human infants; however, the optimal dose will ultimately need to be determined in humans.

Another consideration in human infants is the presence of maternally derived Abs, given their potential to diminish vaccine responses in newborns. To our knowledge, the impact of maternally derived HA stem-specific Abs on newborn stem-based vaccine responses has not been evaluated. Of note, in human newborns administered a gp120 subunit vaccine adjuvanted with MF59, there was no correlation between the level of maternal Ab at birth and the magnitude of the infant Ab response[70]. Stem-specific Abs in humans represent a low proportion of the HA-specific response due to the subdominant nature of the response. We speculate that the low abundance of maternal stem-specific Ab, together with the inclusion of an adjuvant, would constrain inhibition by maternal Abs. Studies wherein the vaccine is administered to newborns that have maternal Abs representing an array of influenza virus/vaccine specificities would provide needed mechanistic insights into the impact of maternal antibodies on the vaccine response. The AGM model employed here will be a valuable setting for these studies, given the similarities between NHP and humans in the placental transfer of Ab.

There are limitations to our study. At present, we lack a fully annotated AGM immunoglobulin genome, which constrains B cell repertoire analysis. Further, viral measurements in the BAL were conducted at a single time point, d7 p.c. Thus, the temporal dynamics of viral clearance in the lung is not fully known. We also note that the method of infection does not fully mimic the normal transmission route of IAV infection. AGMs were administered virus by the combined intranasal and intratracheal routes, which, while ensuring the respiratory tract is exposed to virus, differs from the aerosol route of natural human infection. Finally, our analysis is restricted to the assessment of NK cell-mediated ADCC.

In summary, our findings demonstrate for the first time that delivery of H1ssF + AddaVax can elicit robust broadly reactive Ab responses in newborn nonhuman primates and that these Abs can be protective. Further, we found neutralising and ADCP competent Abs were associated with viral control following challenge. The promising results from this study support the continued exploration of this vaccine for use in newborns and extend our understanding of the effector functions that can provide protection in this vulnerable population.

## Methods

### Experimental design

African green monkeys (vervets) were bred, housed, immunised, and sampled at the Vervet Research Colony at the Wake Forest University School of Medicine. Male adults 5–7 years old (equivalent to ~20–28 human years) received either PBS (Ctrl, $n = 3$) or H1ssF + AddaVax ($n = 3$). Adults received 50 µg of H1ssF[22] with AddaVax (250 µl Invivo-Gen vac-adx-10)) intramuscularly in the deltoid muscle of both left and right arms (100 µg of H1ssF/dose). We determined the vaccine dose based on studies utilising the H1ssF vaccine in adult macaques[24]. Animals were boosted with the same dose at d39 p.v. Plasma samples were collected at d0 p.v., d10 p.v., d39 p.v., d11 p.b., d45 p.b., and d7 p.c. On d45 p.b., adult AGM were challenged with $5 \times 10^7$ TCID$_{50}$ (3.5 mL total) of A/California/07/2009 (Ca09) (2.5 mL given by intratracheal installation, and 0.5 mL in each nostril) (BEI Resources). For newborn studies, AGMs (healthy and above 300 g) were immunised at 3–5 days post birth (equivalent to ~12-20 day old human). Newborn AGM were assigned to vaccine groups, corresponding to PBS or Luciferase mRNA-LNP (non-HA vaccinated controls) (Ctrl) ($n = 5$), H1ssF ($n = 4$), or H1ssF + AddaVax ($n = 8$). Animals were distributed so that there was an equal number of males and females within each vaccine group. Newborns received 20 µg of H1ssF with or without AddaVax (125 µl of AddaVax (InvivoGen)) intramuscularly in the deltoid muscle of both left and right arms (40 µg of H1ssF/dose). Animals were boosted with the same dose at d39/40 p.v. Plasma samples were collected at d1 p.v., d10 p.v., d39/40 p.v., d10/11 p.b., d41/45 p.b., and d7 p.c. On d41/45 p.b., newborn AGM were challenged with $1 \times 10^7$ TCID$_{50}$ (0.75 mL) of A/California/07/2009 (Ca09) (BEI Resources) (0.5 mL given by intratracheal installation, and 0.125 mL in each nostril). The BAL volumes used were 5 ml for newborns and 30 ml for adults. Isoflurane was used as an anaesthetic and was administered as an inhalant (2–5% inhalant) during blood draws. Blood draws and BAL sampling were performed in adults under ketamine at 10 mg/kg. Prior to euthanasia, infants were administered an intramuscular dose of ketamine 10 mg/kg as a sedative, in addition to isoflurane (2–5% inhalant). For euthanasia, an intravenous injection of sodium pentobarbital was administered at a lethal dose of 100 mg/kg. BAL samples were acquired after infants had been euthanized.

### Sex as a biological variable

In the adult study, we used only males due to animal availability. For our newborn study, sex was considered as a biological variable, with equally matched numbers of male and female AGMs. No differences by sex were observed.

### Study approval

The animal care and use protocol was adherent to the US Animal Welfare Act and Regulations. Permission was granted to perform all animal experiments by the Wake Forest University Institutional Animal Care and Use Committee. AGM were housed and cared for in accordance with state, federal, and institute policies in facilities accredited by the American Association for Accreditation of Laboratory Animal Care (AAALAC) under standards established in the Animal Welfare Act and the Guide for the Care and Use of Laboratory Animals.

## C-Reactive protein quantification

C-reactive protein levels in the plasma were assessed at 24 h post vaccination using a human C-Reactive Protein ELISA kit KR9710s (30-9710S) from ALPCO Diagnostics as per the manufacturer's instructions. The plate was read at 450 nm on a Elx800 Absorbance Microplate Reader (BioTek). Amounts were calculated based on the standard curve generated using the reagent provided in the kit.

## ELISA for the Detection of HA NC99 stem-specific Ab

Trimeric A/NewCaledonia/20/1999 (H1N1) stabilised HA stem protein[22] was used to assess stem-specific Ab. For cross-reactivity ELISAs, trimeric recombinant full-length HA proteins were used. The panel included: A/New Caledonia/20/1999 (H1) (BEI Resources, NR-48873), A/California/07/2009 H1(BEI Resources, NR-42635), PR8 (H1) (BEI Resources, NR-19240), A/Singapore/1/1957 (H2) (BEI Resources, NR-52249), A/Wisconsin/67/2005 (H3) (BEI Resources, NR-49237), A/Vietnam/1204/2004 (H5) (VRC, NIH), A/Anhui/1/2013 (H7) (BEI Resources, NR-44365), and A/HongKong/33982/2009 (H9) (BEI Resources, NR-41792). We used FI6V3 HA-stem mAb as the positive control at 0.156 μg/mL (VRC, NIH)[32]. We utilised HA ectodomain glycan mutants on the stem to assess the epitope specificity of the stem response. Q27NHA2 and N29THA2 mutations were made to introduce a glycan at 27HA2 (H1-N27) to block the anchor epitope[26]. I45NHA2 and G47T mutations were made to introduce a glycan at 45HA2 (H1-N45) to block the central epitope[33]. The H1-N45/27 contained all four mutations[26]. Full length HA ectodomain from A/NewCaledonia/20/1999 (H1N1) was used as the H1-WT control[26]. To assess Ab production, ELISA microplates (96-half well) (Greiner bio-one) were coated with 40 ng/well of antigen (NC99 trimeric stem, H1-N27, H1-N45, H1-N45/27, H1-WT) or 100 ng/well of recombinant HA protein (cross reactivity ELISA) in phosphate buffered saline (PBS) (stem and epitope specificity ELISA) or sodium carbonate/bicarbonate coating buffer (pH 9.5) (cross-reactivity ELISA) overnight at 4 °C. The plates were blocked with 1X blocking buffer (10X Casein Blocking Buffer, Sigma-Aldrich) plus 2% goat serum (Lampire Biologicals) for 1 h and then washed. Wash buffer contained PBS with 0.1% Tween 20. Plasma samples were serially diluted in 1X blocking buffer. Wells that contained no antigen served as a negative control. HRP-conjugated antibodies specific for goat anti-monkey IgM (Fitzgerald #43R-IG074hrp) (1:10,000) and goat anti-monkey IgG (Fitzgerald #43C-CB1603) (1:5,000) were used to detect bound antibodies. For detection of IgA, goat anti-monkey IgA-Biotin (Fitzgerald 43R-IG002bt) (1:5,000) was used, followed by NeutrAvidin-HRP (Thermo Fisher 31001) (1:5,000). Plates were developed using 25 μl of 3, 3′, 5, 5′-Tetramethylbenzidine dihydrochloride (TMB) (Sigma-Aldrich), and stopped after 30 mins with 25 μl of 2 N $H_2SO_4$. Plates were read at 450 nm on an Elx800 Absorbance Microplate Reader (BioTek) using Gen 5 software (BioTek). For each dilution, the $OD_{450}$ from the no antigen wells was subtracted from the stem-coated wells. Threshold titre was defined as the value that reached 3X the assay background, i.e., wells that only received 1X blocking buffer + stabilised stem (no sample). To evaluate the binding pattern across HA molecules, for each animal, the ratio of HA TT: NC99 HA TT was calculated. AUC was calculated with GraphPad Prism software using the trapezoidal method. The values used included: x axis = plasma dilutions, and y axis = absorbance 450 nm. The plasma dilutions used were the nontransformed values.

## ELISA for the Detection of Monkey Anti-Nuclear Abs

Monkey anti-nuclear IgM (# 670-115-ANM) and IgG (# 670-110-ANM) (Alpha Diagnostic International) was detected as per manufacturer's instructions. Plasma samples were diluted at 1:50. The monkey ANA IgM and IgG ELISA kits is based on the binding of monkey Abs to extractable nuclear antigens (ENA). ENA antigen in these kits is a combination of several nuclear autoantigens (dsDNA, SSA/Ro, SSB/LA, Scl70, Sm, RNP, and Jo-1). Plates were read at 450 nm on a Elx800

Absorbance Microplate Reader (BioTek) running Gen 5 software (BioTek). The dotted line represents the limit of detection of the assay (blank).

## ELISA for PR8-Specific IgG

ELISA microplates (96-half well) (Greiner bio-one) were coated with 14 μg/mL of A/PuertoRico/8/1934 (H1N1) (Charles River) in sodium carbonate/bicarbonate coating buffer (pH 9.5) (25 μl, 0.35 μg per well) overnight at 4 °C. Plates were blocked with 1X Blocking Buffer (10X Casein Blocking Buffer, Sigma-Aldrich) with 2% goat serum (Lampire Biologicals) for 1 h and then washed with PBS + 0.1% Tween 20. Plasma samples were serially diluted in 1X blocking buffer. Wells that contained no virus served as a negative control. Goat anti-monkey IgG (Fitzgerald #43C-CB1603) (1:5,000) was used to detect bound IgG. Plates were developed using TMB (Sigma-Aldrich) and stopped using 2 N $H_2SO_4$ after 30 mins. Plates were read at 450 nm on an Elx800 Absorbance Microplate Reader (BioTek) running Gen 5 software (BioTek). For each dilution, the OD from the non-virus-coated wells was subtracted from the virus-coated wells. The threshold titre was defined as the value that reached 3X the assay background, i.e., wells that only received 1X blocking buffer + virus (no sample).

## ELISA for the detection of Fluzone-Specific IgG

ELISA microplates (96-half well) (Greiner bio-one) were coated with 4 μg/mL of Fluzone High Dose Quad 2023-2024 formula produced by Sanofi Pasteur in sodium carbonate/bicarbonate coating buffer (pH 9.5) (25 μl, 100 ng per well) overnight at 4 °C. Plates were blocked with 1X Blocking Buffer (10X Casein Blocking Buffer, Sigma-Aldrich) with 2% goat serum (Lampire Biologicals) for 1 h and then washed with PBS containing 0.1% Tween 20. Plasma samples were serially diluted in 1X blocking buffer. Wells that contained no virus served as a negative control. Goat anti-monkey IgG (Fitzgerald #43C-CB1603) (1:5,000) was used to detect bound IgG. Plates were developed using TMB (Sigma-Aldrich) and stopped using 2 N $H_2SO_4$ after 30 mins. Plates were read at 450 nm on an Elx800 Absorbance Microplate Reader (BioTek) running Gen 5 software (BioTek). For each dilution, the $OD_{450}$ from the non-Fluzone-coated wells was subtracted from the Fluzone-coated wells. The threshold titre was defined as the value that reached 3X the assay background, i.e., wells that only received 1X blocking buffer + virus (no sample).

## NaSCN dissociation assay for determining NC99 stem-specific avidity

ELISA microplates (96-half well) (Greiner bio-one) were coated with 40 ng/well of NC99 trimeric stem in phosphate buffered saline (PBS) overnight at 4 °C. The plates were blocked with 1X blocking buffer (10X Casein Blocking Buffer, Sigma-Aldrich) plus 2% goat serum (Lampire Biologicals) for 1 h and then washed. Wash buffer contained PBS with 0.1% Tween 20. Plasma samples were serially diluted in 1X blocking buffer and incubated for 2 hrs. Wells that contained no antigen served as a negative control. NaSCN was diluted in 2-fold dilutions (starting dilution=5 M) and added to the plate for 15 mins prior to washing the plate. HRP-conjugated Ab specific for goat anti-monkey IgG (Fitzgerald #43C-CB1603) (1:5,000) was used to detect bound antibodies. Plates were developed using TMB (Sigma-Aldrich) and stopped after 30 mins with 25 μl of 2 N $H_2SO_4$. Plates were read at 450 nm on a BioTek Elx800 Absorbance Microplate Reader running Gen 5 software (BioTek). For each dilution, the OD from the non-stem-coated wells was subtracted from the stem-coated wells. NC99 stem-specific IgG avidity was measured by determining the NaSCN $IC_{50}$ concentration that resulted in a 50% reduction in optical absorbance compared to the untreated sample.

## RT-PCR for Viral detection

Viral RNA was extracted from the bronchoalveolar lavage samples using a Quick-RNA Viral 96 Kit (Zymo Research). For cDNA synthesis

and viral RNA quantification, the LUNA Universal Probe One-Step RT-qPCR Kit (New England BioLabs) was used along with primers designed by the CDC to capture the Influenza A virus M2 gene (see primer sequences Supplementary Table 3). These primers were designed as part of a multiplex assay to quantify SARS-CoV-2, Influenza B, and Influenza A viruses. For this study, only those primers specific for Influenza A viruses were used. Primers and probes of the CDC recommended sequence (see below) were purchased from Integrated DNA Technologies (IDT). The probe contained the reporter molecule 6-carboxyfluorescein (FAM) at the 5' end with an internal ZEN™ quencher and the Iowa Black FQ quencher (IABkFQ) at the 3' end. The one-step RT-qPCR reaction was performed in a volume of 10 μl with 0.25 μM FAM-labelled probe, 0.45 μM each forward primer, 0.65 μM InfA Rev1 and 0.2 μM InfA Rev2 primers. Quantitative RT-PCR (qRT-PCR) was performed using a BioRad CFX384 Real-Time System, accompanied by BioRad CFX Maestro software for analysis. Viral genomes/mL (vg/mL) were calculated on the basis of a standard curve generated by using a synthetic DNA construct (purchased from IDT) with the target sequences for the Influenza A virus (Genbank accession number MN976418.1, A/Hawaii/66/2019[H1N1], segment 7, matrix gene). The sequence for the DNA construct used for standard curve generation is indicated below where uppercase symbols identify the influenza A and influenza B targets. The total vg/mL for the sample was calculated on the basis of the amount present in the sample volume used for the viral RNA extraction (200 μl) followed by adjustment to total vg/ml. The CDC designed primer sequences used for RT-qPCR and the Synthetic DNA target sequence are provided in Supplementary Table 4.

## IL-6 Quantification
IL-6 was quantified using a human inflammatory cytokine bead array (BD Biosciences) performed per the manufacturer's instructions on BAL collected d7 p.c. Samples were acquired on a BD Fortessa X20 (BD Biosciences) and data analysed using FCAP Array software.

## Microneutralization assay
Generation of the replication-restricted reporter (R3ΔPB1) virus H1N1 subtypes (A/California/07/2009 & A/New Caledonia/20/1999) as well as Rewired R3ΔPB1 (R4ΔPB1) virus H5N1 (A/Vietnam/1203/04) is described elsewhere[40]. Briefly, to generate the R3/R4ΔPB1 viruses, the viral genomic RNA encoding functional PB1 was replaced with a gene encoding the fluorescent protein (TdKatushka2), and the R3/R4ΔPB1 viruses were rescued by reverse genetics and propagated in the complementary cell line which expresses PB1 constitutively. Each R3/R4ΔPB1 virus stock was titrated by determining the fluorescent units per mL (FU/mL) prior to use in the experiments. For virus titration, serial dilutions of virus stock in OptiMEM were mixed with pre-washed MDCK-SIAT1-PB1 cells ($8 \times 10^5$ cells/mL) and incubated in a 384-well plate in quadruplicate (25 μL/well). Plates were incubated for 18–26 h at 37 °C with a 5% $CO_2$ humidified atmosphere. After incubation, fluorescent cells were imaged and counted by using a Celigo Image Cytometer (Nexcelom) with a customised red filter for detecting TdKatushka2 fluorescence.

For the microneutralization assay, serial dilutions of RDE (Receptor Destroying Enzyme) treated plasma were prepared in OptiMEM and mixed with an equal volume of R3/R4ΔPB1 virus (~$8 \times 10^4$ FU/mL) in OptiMEM. After incubation at 37 °C and 5% $CO_2$ humidified atmosphere for 1 h, pre-washed MDCK-SIAT1-PB1 cells ($8 \times 10^5$ cells/well) were added to the plasma-virus mixtures and transferred to 384-well plates in quadruplicate (25 μL/well). Plates were incubated and counted as described above. Target virus control range for this assay is 500 to 2,000 FU per well, and a cell-only control is acceptable up to 30 FU per well. The percent neutralisation was calculated for each well by constraining the virus control (virus plus cells) as 0% neutralisation and the cell-only control (no virus) as 100%

neutralisation. A 7-point neutralisation curve was plotted against serum dilution for each sample, and a four-parameter nonlinear fit was generated using Prism (GraphPad) to calculate the 80% ($IC_{80}$) inhibitory concentrations.

## Antibody-dependent cellular cytotoxicity
KHYG-1 cells transduced with rhesus macaque CD16 (kind gift of Dr. David Evans)[71] were used to perform the ADCC assay adapted from ref. 72. ELISA plates (Corning) were coated with 50 μL (2 μg/mL in PBS) of recombinant HA protein (A/California/07/2009) (BEI Resources) and left to incubate at 4 °C overnight. Plates were washed and then blocked with PBS + 5% bovine serum albumin (BSA) at 37 °C in a 5% $CO_2$ incubator for 30 mins. Blocking solution was removed, and heat-inactivated (HI) plasma samples were serially diluted (starting dilution 1:10) and incubated at 37 °C in a 5% $CO_2$ incubator for 2 hrs. Following washing, $1 \times 10^5$ cells KHYG-1 MmCD16 cells were added per well (50 μL) in media (RPMI + 10% FBS + 1% L-glut, 0.2% Primocin). Recombinant human IL-2 (Biolegend, #589104) (10 U/mL, 50 μg/mL) was added to cells. Anti-CD107a-PE (Biolegend, #328608) (1:50) was added to 25 μL of warmed media and then added to plates containing cells. Golgi Plug (1:1000) and Golgi Stop (1:1500) (BD Biosciences) was added to cells in 25 μL of warmed media. Cells (100 μL total) were incubated at 37 °C in a 5% $CO^2$ incubator for 5 hrs. Cells were washed with PBS and stained with Zombie Violet (Biolegend) for 15 mins at room temperature (RT) in the dark. Cells were washed with FMF and fixed with 2% paraformaldehyde. ADCC was assessed using flow cytometry on a BD X20 Fortessa running FACS Diva software V9 (Becton Dickinson). The percentage of CD107a + NK cells was calculated by subtracting the PBS from the HA values for each dilution.

## Antibody-dependent complement deposition
Antibody-dependent complement deposition of HA-coated beads was adapted from ref. 53. HA-coated 1 μm non-fluorescent NeutrAvidin beads were prepared as in the ADCP assay and incubated with diluted, heat-inactivated plasma samples (2-fold dilutions, starting dilution 1:5) for 2 hrs at 37 °C. Lyophilised guinea pig complement (Cedarlane, CL4051) was reconstituted in ice-cold $dH_2O$ and diluted in 1:60 veronal buffer with 0.1% gelatin (GVB + + Boston BioProducts, #IBB-300X). Next, 150 μl of the diluted complement was added to the opsonized beads and incubated for 20 minutes at 37 °C. Beads were washed with 15 mM EDTA and stained with anti-guinea pig C3-FITC (MP Biomedicals, # 855385). Samples were washed, and ADCD was assessed using flow cytometry on a BD X20 Fortessa (BD Biosciences) using FACS Diva software V9 (Becton Dickinson). Complement scores were calculated using the percentage of C3+ beads×GMFI/1000 at 1:5 dilution.

## Antibody-dependent cellular phagocytosis
THP-1 phagocytosis of HA-coated beads was adapted from ref. 73. Biotinylated HA A/California/07/2009 (kindly provided by the DCVC, Duke University) was adsorbed onto 1.0 μm of fluorescently labelled beads called FluoSpheres NeutrAvidin (yellow-green fluorescent (505/515)) at a ratio of 10 μg protein to 10 μl beads and left to incubate overnight at 4 °C. 10 μl of HA-coated beads was incubated with 10 μl of heat inactivated plasma diluted in PBS in a 96-round bottom plate (2-fold dilutions, starting dilution 1:800) for 2 hrs at 37 °C. Any unbound Ab was washed away, and THP-1 cells (RPMI + 10% FBS + 1% L-glut, 1% P/S) ($5 \times 10^4$ cells/well) added and incubated for 16 hrs at 37 °C. Cells were fixed and phagocytosis assessed using flow cytometry on a BD X20 Fortessa (BD Biosciences) running FACS Diva software V9 (Becton Dickinson). Phagocytosis scores were determined by the bead-positive cells×the geometric mean/1000 at 1:800 dilution.

## Statistical analysis
Several analytic methods were used to evaluate the data. For analyses that examined the impact of different treatments over time, two-way

repeated measures ANOVA testing was performed. If there was evidence of overall treatment effects, then comparisons between groups were made at each time point, adjusted for multiple comparisons using a Tukey's approach. For analyses comparing treatments at only one time point, one-way ANOVA models were used. If there was evidence of overall treatment effects, pairwise comparisons were performed using Tukey's or Fisher's Least Significant Difference approach for handling multiple comparisons. For analyses that were examining the associations between two measures on a continuous scale, correlations were calculated and examined. Finally, for endpoints that were not able to be compared using parametric approaches (i.e., using t-tests or ANOVA models), Wilcoxon rank-sum or Kruskal-Wallis tests were performed. Depending on the analyses performed, the following software packages were used Prism 10.1.2 (GraphPad, La Jolla, CA, USA) or SAS Version 9.4.

### Reporting summary

Further information on research design is available in the Nature Portfolio Reporting Summary linked to this article.

## Data availability

All data are included in the Supplementary Information or available from the authors, as are unique reagents used in this Article. Source data are provided in the Source Data file. Source data are provided in this paper.

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

## Acknowledgements

We thank the Wake Forest Animal Resources Programme and the veterinary and technical staff of the Vervet Research colony for the care of animals and assistance with animal procedures. We thank Malaysia Janta Carter for help with acquiring a portion of the RT-PCR infection data. We thank the Duke University DCVC for the provision of biotinylated HA A/California/07/2009 and Dr. David Evans for KHYG-1 cells transduced with rhesus macaque CD16. The following reagents were obtained through BEI Resources, NIAID, and NIH: H1 Hemagglutinin (HA)

Protein with C-Terminal Histidine Tag from Influenza Virus, A/New Caledonia/20/1999 (H1N1), Recombinant from Baculovirus, NR-48873. H1 Hemagglutinin (HA) Protein with C-Terminal Histidine Tag from Influenza Virus, A/California/07/2009 (H1N1)pdm09, Recombinant from Baculovirus, NR-42635. H1 Hemagglutinin (HA) Protein with C-Terminal Histidine Tag from Influenza Virus, A/Puerto Rico/8/1934 (H1N1), Recombinant from Baculovirus, NR-19240. H2 Hemagglutinin (HA) Protein from Influenza A Virus, A/Singapore/1/1957 (H2N2), Recombinant from Baculovirus, NR-52249. H3 Hemagglutinin (HA) Protein with C-Terminal Histidine Tag from Influenza Virus, A/Wisconsin/67/2005 (H3N2), Recombinant from Baculovirus, NR-49237. H7 Hemagglutinin (HA) Protein with C-Terminal Histidine Tag from Influenza Virus, A/Anhui/1/2013 (H7N9), Recombinant from Baculovirus, NR-44365. Hemagglutinin (HA) Protein from Influenza Virus, A/ Hong Kong/33982/2009 (H9N2), Recombinant from Baculovirus, NR-41792. Influenza A Virus, A/California/07/2009 (H1N1) pdm09, Egg Isolate (Produced in Eggs), NR-13663. Figures 1A and 3A were created with BioRender.

## Author contributions

Conceptualisation: K.F.C., M.K. and M.A.A.-M. Methodology: K.F.C., B.C.H., C.L.P., R.A.G. and D.AO. Investigation: K.F.C., B.C.H., C.L.P., R.A.G., R.B.D., M.S. and D.A.O. Visualisation: K.F.C. and M.A.A.-M. Funding acquisition: M.A.A.-M. Project administration: M.A.A.-M. upervision: M.A.A.-M and M.K. Writing – original draft: K.F.C. and M.A.A.-M. Writing – review & editing: K.F.C., B.H., C.L.P., R.A.G., R.B.D., D.A.O., M.K. and M.A.A.-M.

## Funding

National Institute of Allergy and Infectious Diseases R01 AI146059 (MAA-M) and T32AI007401 (MAA-M); National Cancer Institute Cancer Centre Support Grant P30CA012197 (R. Mesa). The Vervet Research Colony is funded in part by P40 OD010965 (M. Jorgensen). This study was supported in part by the Vaccine Research Centre, an intramural Division of the National Institute of Allergy and Infectious Diseases, National Institutes of Health (M.K.).

## Competing interests

M.K. is listed as inventor of patents and patent applications on vaccine immunogens used in this study filed by the U.S. Department of Health and Human Services. The remaining authors declare no competing interests.
