## [Transparent Peer Review file · Nature Communications]

Antibody Function Predicts Viral Control in Newborn Monkeys Immunized with an Influenza Virus HA Stem Nanoparticle

Corresponding Author: Dr Martha Alexander-Miller

Version 0:

Reviewer comments:

Reviewer #1

(Remarks to the Author)

This manuscript describes an important piece of work that assesses the immunogenicity and efficacy of a HA stem nanoparticle that has had improved immunogenicity over that of traditional flu vaccines in small animals and adult NHPs/humans in infants, a vulnerable population for flu that currently has no vaccine product available until 6 mo of age. I have a few comments to improve the impact of the work:

1. More safety data would be of interest for the translation to human infants - what was the wt gain trajectory of the vaccinated infants compared to the colony or unvaccinated infants, especially in the adjuvant vs no adjuvant group? What were the injection site observations?
2. The manuscript would benefit from a figure comparing the vaccine-elicited responses of infants and adults. It is not surprising that this protein-based antigen vaccine elicits improved antibody responses and functions given results from Hep B, COVID vaccine studies in humans and HIV vaccine studies in NHP infants - the text should be modified accordingly.
3. This vaccination is performed in infants without the presence of maternal antibodies - their potential impact and outlining the studies that are needed to determine the impact should be added to the discussion (also how matAbs impact current flu vaccines in <6mo in prior studies should be added to intro)

Minor

1. can the authors comment on the ability to perform B cell repertoire studies in this model? these would be useful in aiming to boost the responses elicited in early life for persistent protection
2. Make it clear that the "prolonged period of infancy" of NHPs is compared to small animal models, not humans
3. Detail how AUC for binding antibodies was calculated, these seem very high, but may be related to log transformation vs not. Are ID50s not able to be calculated? Showing the binding curves in supplemental could be helpful
4. with the NC99 HA mutants, can you add the binding level to the WT HA to show the change in binding?
5. Add what route the animals were challenged in results and methods
6. Page 9 - the level of abs and viral load cannot be "correlated" statistically with only 3 animals. Refine the language to just describe the VL in the animals with "high" vs "low" abs. Change language also in discussion
7. can the "rapid recall response" upon challenge be related to results from other animal models?
8. Fig 2 - can the protective level of neutralization from human or NHP studies be added to the graphs? What is the goal for heterologous virus neutralization in plasma?

9. add what the needs are for dosing studies to optimize human infant clinical trials - can this be done in AGMs?
10. can you measure IgA responses in plasma to the vaccine - are they measurable? Does plasma IgG correlate with BAL IgG (and how is BAL IgG levels accounted for volume)
11. add r values to text of statical correlations (in addition to the p value)
12. the improved functional responses (and Fc and glycan profiles) in infants vs adults has also been studied in COVID vaccines and HIV vaccines - these would be good literature to point to. This manuscript gives the impression that infants are still thought of as "poor responders" when it comes to B cell responses to vaccines, and that is panning out to be untrue in many settings, especially with protein-based antigen vaccines
13. Could fine HA specificity of the responses determine non-neut function? This should be included in addition to subclass, FcR binding, and glycans

Reviewer #2

(Remarks to the Author)

In the manuscript entitled "Antibody Function Predicts Viral Control in Newborn Monkeys Immunized with an Influenza Virus HA Stem Nanoparticle" Crofts K.F et al. examined immunogenicity of H1ssF nanoparticle vaccine with AddaVax adjuvant in NHP model. The authors first tested H1ssF+AddaVax vaccine in adult monkeys to confirm H1ssF vaccine's immunogenicity and protective efficacy. Although they saw good anti-stem binding antibody in H1ssF+AddaVax group but they didn't show neither neutralizing antibody nor cellular immune responses after two i.m. immunizations. Then the authors tested H1ssF vaccine in newborn monkeys. They observed good binding antibodies against HA stem with some difference in quality. Interestingly, the authors found that neutralizing Abs and ADCP activity are important contributors to Ab-mediated influenza virus control in newborn monkeys. The overall study is well performed, and conclusions are supported by results. Below are my concerns and suggestions.

Major points:

1. What would make this work even more compelling is if anti-stem antibodies were given before neonatal immunization to mimic a setting more faithfully representing a real-life.
2. Figs. 1C & 3C symbol legends: Please remove symbols as they are confusing. I was looking for red triangles and circles. Just show colors in line will be more appropriate. Also please change one red square series to red circles in Fig. 1C as there are two red square lines.
3. p.7 line 4: "robust generation of stem-specific IgG at d10 p.v. in vaccinated adults": Have the authors seen similar pattern in mice or other models? Could be the sensitivity issue of anti-monkey IgM antibody used?
4. Please discuss about the lack of a defined regulatory pathway for approval of non-HA influenza vaccines.
5. p. 25 line 17: This group might have anti-AGM IgG subclass antibodies. 'Lara A. et al., Peripheral immune response in the African green monkey model following Nipah-Malaysia virus exposure by intermediate-size particle aerosol. PLoS Negl Trop Dis. 2019 Jun; 13(6): e0007454. doi: 10.1371/journal.pntd.0007454'.
6. Can the authors add comments about safety of H1ssF+AddVax vaccine in neonates?

Minor points:

1. "ELISA for the Detection of PR8-Specific IgG" section; "14 µg/well" should be "14 µg/mL". Also please confirm the OD. If you don't stop TMB reaction OD should be 650nm.
2. Please ensure correct formatting (sub/superscript) of CO₂, 10⁵ cells, IC80 etc.
3. Please ensure consistent description for "AddaVax". AddaVax, Addavax etc.
4. Please ensure correct description for "A/California/07/2009" not "A/California/07/09".
5. p.16 line 8: There is no Supplementary Fig. 5a. Should be Fig. 5?
6. Fig. 4F: "PBS" should be "Ctrl" for consistency.

Reviewer #3

(Remarks to the Author)

The authors tested an HAstem based vaccine with Addavax (an MF59-like adjuvant) in juvenile african green monkeys for breadth of antibody responses and reduction of viral titers in BALs.

The vaccine elicited antibodies that neutralized an H1N1 and H5N1 historical viruses.

Overall, the study is well performed.

Several studies have used HAstem vaccines, but this is the first study to examine vaccine effectiveness in young monkeys.

To determine the effectiveness of antibodies elicited by HAstem, the study would be stronger if the antibodies were tested against modern group 1 influenza vaccines, i.e. 2023-2024 season to show that the elicited antibodies protect against drifted H1N1 variants.

Version 1:

Reviewer comments:

Reviewer #1

(Remarks to the Author)

Overall, the authors performed a comprehensive response and editing to the reviewers comments, which has improved the quality and translational potential of the findings.

My only recommendation is to include the injection site observations, even though negative, to the results of the manuscript and to add the limitation of the AGM model for studying B cell repertoires in the discussion.

Reviewer #2

(Remarks to the Author)

All comments have been addressed to the best of the authors ability. I have no further comment.

Reviewer #3

(Remarks to the Author)

NCOMMS-24-35975: Antibody Function Predicts Viral Control in Newborn Monkeys Immunized with an Influenza Virus HA Stem Nanoparticle

We appreciate the reviewer's interest in our study and their positive comments on its importance. This study was designed to determine whether a universal influenza vaccine targeting the stem region of HA could be effective in newborns. This ferritin-based vaccine platform has shown promise in adult humans; therefore, if approved, young infants would be a key target population. Evaluating its performance in this age group is critical. The immune system of young infants has substantial differences in wiring and repertoire compared to adults. Thus, whether this universal influenza vaccine approach would be effective was unknown and necessitated evaluation. The newborn NHP is inarguably the best pre-clinical model for addressing this question. A concern raised by the reviewers was the absence of studies that address the role of maternal antibody in the infant response to vaccination. This is clearly an important question worthy of study. However, it is a separate and complex question that is out of the scope of the current study, which represents a significant undertaking given the model employed. A multi-year study to address maternal antibody requires multiple animals and additional funding. In addition to the novel information regarding the ability of this vaccine approach to work in young infants, we believe the manuscript makes a substantial contribution by revealing antibody attributes that are associated with clearance of influenza virus in this age group and the comparative response between newborns and adults. No studies in the context of a universal flu vaccine have been previously performed and such insights are essential for the further development of this approach. The results reported in our study significantly increase our understanding of this important area and move the field forward in a meaningful way.

We sincerely appreciate the opportunity to respond to the insightful comments from the reviewers that have significantly improved our manuscript. We have carefully addressed each comment and have performed additional experiments as suggested by the reviewers that are now included. We have added 17 new panels to the paper (Fig. S1A-D; Fig. S4C; Fig. S5B-E; Fig. S6C-D; Fig. S8; Fig. S9A-E). Our detailed response to the reviewers comments follows.

Reviewer #1

Overall comments: This manuscript describes an important piece of work that assesses the immunogenicity and efficacy of a HA stem nanoparticle that has had improved immunogenicity over that of traditional flu vaccines in small animals and adult NHPs/humans infants, a vulnerable population for flu that currently has no vaccine product available until 6 mo of age.

We thank the reviewer for this positive comment regarding the importance of our study.

I have a few comments to improve the impact of the work:

- 1. More safety data would be of interest for the translation to human infants - what was the wt gain trajectory of the vaccinated infants compared to the colony or unvaccinated infants, especially in the adjuvant vs no adjuvant group?**

We appreciate the reviewer's suggestion for including additional safety data in the manuscript. In addition to the data in Fig. S3 that show similar CRP and temperature in the H1ssF+AddaVax animals compared to the Ctrl/H1ssF animals, we now include a time course of each infant's weight in (Fig. S4C). No significant difference in weight was observed among the different vaccine groups. We have addressed this in **lines 238-240** in the manuscript.

What were the injection site observations?

Following vaccination, infants were monitored for 15 mins prior to return to their mothers. No immediate reaction was noted. Further, the infant and mother pair were housed in a single cage for 24h before return to the colony. At the 24h p.v. blood draw, infants were assessed for changes at the vaccination site as well as general health characteristics (i.e. alert, latching on to mothers). No

vaccine-induced changes were observed.

- 2. The manuscript would benefit from a figure comparing the vaccine-elicited responses of infants and adults. It is not surprising that this protein-based antigen vaccine elicits improved antibody responses and functions given results from Hep B, COVID vaccine studies in humans and HIV vaccine studies in NHP infants - the text should be modified accordingly.** We thank the reviewer for this suggestion. We have included the requested data in figures that compare adult and newborn Ab quantity and quality (Fig. S1 and Fig. S9). We have discussed these findings in **lines 145, 246-248, and 462-482** in the manuscript. We have noted the results with other protein vaccines in newborns/young infants within this section in **lines 468-474, and 478-482**.
- 3. This vaccination is performed in infants without the presence of maternal antibodies - their potential impact and outlining the studies that are needed to determine the impact should be added to the discussion (also how matAbs impact current flu vaccines in <6mo in prior studies should be added to intro).**

As suggested, we have included information in the introduction regarding how maternal antibodies impact current influenza vaccines in infants under 6 months of age (**lines 65-75**). As suggested, we have incorporated a discussion around maternal antibody and this vaccine moving forward (**lines 508-521**).

Minor:

- 1. Can the authors comment on the ability to perform B cell repertoire studies in this model? these would be useful in aiming to boost the responses elicited in early life for persistent protection.** We agree that B cell repertoire analysis would be of interest. At present, in contrast to macaques, we lack a fully annotated AGM immunoglobulin genome. This limitation constrains our ability to perform detailed repertoire studies in AGMs at present.
- 2. Make it clear that the "prolonged period of infancy" of NHPs is compared to small animal models, not humans.** We appreciate the reviewer's comment to clarify this statement. We have modified the text accordingly in **line 107**.
- 3. Detail how AUC for binding antibodies was calculated, these seem very high, but may be related to log transformation vs not. Are ID50s not able to be calculated? Showing the binding curves in supplemental could be helpful:** AUC was calculated with Prism software using the trapezoidal method. The values used included: x axis = plasma dilutions, and y axis = absorbance 450 nm. The plasma dilutions used were the untransformed values. We have added this information to the methods (**line 618**) and included the binding curves in the Fig. S5 for increased clarity.
- 4. With the NC99 HA mutants, can you add the binding level to the WT HA to show the change in binding?** We agree that this is helpful information. The WT HA AUC is included in the supplemental figures along with the stem mutants. Data for adults can be found in **line 167** and Fig. S2 and newborns in **lines 283, 286** and Fig. S5A.
- 5. Add what route the animals were challenged in results and methods.** We thank the reviewer for their careful review and apologize for this omission. We have added the challenge routes to **lines 547-549** for adults and **lines 558-560** for newborns in the Materials and Methods section.

- 6. Page 9 - the level of Abs and viral load cannot be "correlated" statistically with only 3 animals. Refine the language to just describe the VL in the animals with "high" vs "low" abs. Change language also in discussion.** We thank the reviewer for this comment, as in the absence of a correlation analysis, an implication of statistical correlation is not warranted. We have corrected the language in the results section (**lines 191-193**) and **lines 392-393** in the discussion to state that the vaccine promoted enhanced stem IgG and reduced viral clearance.
- 7. Can the "rapid recall response" upon challenge be related to results from other animal models?** We now include data describing others studies that have found early recall responses. Although this timepoint is not often assessed, changes in Ab at d6-7 has been seen in mice using a split influenza A (H3N2) virus (quantity) or a COBRA-based influenza vaccine (subclass). Further, robust increases in nAb can be detected at 1 week following challenge in ferrets vaccinated with IBV LAIV. Finally, in humans there is evidence of rapid recall responses following H1ssF vaccination as plasmablasts peak in circulation between d5 and d7 p.v. These early responses align with our findings that adjuvanted H1ssF vaccinated newborns with no detectable nAb to H5N1 prior to challenge exhibit rapid production of these Abs by d7p.c. Neutralizing antibodies were not detectable in ctrl animals, adding additional strength to the conclusion of a rapid recall response. We have a discussion of these findings in **lines 425-430**.
- 8. Fig 2 - can the protective level of neutralization from human or NHP studies be added to the graphs? What is the goal for heterologous virus neutralization in plasma?** At present, the only assay for which there is an established correlate of protection is HAI, with protective titers being defined as those $\geq 1:40$ for adults and $\geq 1:110$ for young children. Unfortunately, the HAI assay is less relevant for stem antibodies than HA head antibodies as this assay relies on antibodies that bind the HA receptor binding site on the head. Stem-specific Abs have been associated with protection, functioning through both neutralizing Ab and non-neutralizing activities. The H1ssF vaccine studied here has been shown to promote broad group 1 reactivity in adults; however, how effective this platform is in eliciting these Abs in newborns was unexplored until the current studies. Broadly neutralizing Ab measured in plasma should reflect that which enters the lung. The development of a broadly protective universal vaccine is a timely goal given the emerging threat of H5N1.
- 9. add what the needs are for dosing studies to optimize human infant clinical trials - can this be done in AGMs?** In the first clinical trial carried out with this vaccine, the HA content of H1ssF was matched to commercial vaccines. Roughly, the 20 μg and 60 μg of HA-ferritin used in the trial contained 15 μg and 45 μg of HA, respectively, that equate to the standard dose in the seasonal vaccine (15 μg per HA) and Flublok (45 μg per HA). The subsequent stem-ferritin trials used the 20 and 60 μg (total protein) to match what was used in the first-in-human ferritin trial. Historically, infants receive smaller doses compared to adults, for the current influenza vaccine it is half of the standard dose. The standard human dose is typically used in adult NHP studies. We expect newborn AGM studies can provide useful information to design initial trials in human infants. However, the optimal dose will ultimately need to be determined in humans. We have discussed this in **lines 501-507**.
- 10. Can you measure IgA responses in plasma to the vaccine - are they measurable? Does plasma IgG correlate with BAL IgG (and how is BAL IgG levels accounted for volume).** We thank the reviewer for this important suggestion. We had not previously measured IgA as typically protein based vaccines primarily induce IgG responses in naïve animals when

administered i.m. We have now performed this analysis, finding stem-specific IgA is generated following vaccination (Fig. S1). These data are of note as we have tested a range of different IAV protein-based vaccines as well as an mRNA-LNP vaccine in the newborn AGM model in our lab and this is the first time we have observed an increase in Ag-specific IgA. Intriguingly, the level of IgA generated was significantly lower in adults compared to newborns. These findings are described in **lines 145 and 246-248** in the results and **lines 466-469** in the discussion. We performed a correlation analysis for IgG and IgA in the plasma of H1ssF+AddaVax adult and newborn animals finding a strong positive correlation (Fig. S1D). We also performed a correlation analysis to understand whether the level of NC99-stem specific IgG in the plasma correlated with the level of NC99-stem specific IgG in the BAL. Although we see significant increases in NC99-stem specific IgG in both plasma and BAL of the H1ssF+AddaVax animals compared to the ctrl and the H1ssF animals, these levels are not significantly correlated at d7 p.c.

With regard to volume as it relates to stem-specific IgG measurement in the BAL, we administer the same volume of PBS to all infants. Thus, even if the recovered volume differs, the same “dilution” of the antibody in the lung occurred and thus titers in the ELISA can be directly compared among animals.

- 11. add r values to text of statistical correlations (in addition to the p value).** We have added the r values to the text as requested.
- 12. The improved functional responses (and Fc and glycan profiles) in infants vs adults has also been studied in COVID vaccines and HIV vaccines - these would be good literature to point to. This manuscript gives the impression that infants are still thought of as "poor responders" when it comes to B cell responses to vaccines, and that is panning out to be untrue in many settings, especially with protein-based antigen vaccines.** -We appreciate this suggestion. We have carefully evaluated the manuscript in an effort to not overstate newborns as generally poor vaccine responders, focusing on what is known about their immune system in general and response to the influenza vaccine. We have added a discussion of examples of studies with other vaccines in **lines 469-482**. Additionally, new figures Fig. S1 and Fig. S9 comparing newborn and adult responses shed light on this issue.
- 13. Could fine HA specificity of the responses determine non-neut function? This should be included** –We performed a correlation analysis for the ADCP activity at d41/45 p.b. and Ab specific for the central epitope, finding no significant correlation ($r = 0.54$, $p = 0.17$). We have included this graph in the supplementary materials (Fig. S8). However, we appreciate that we cannot rule out differences in Abs recognizing other epitopes. This finding is discussed at **lines 365-369** in the manuscript.

Reviewer #2

Overall comments: In the manuscript entitled “Antibody Function Predicts Viral Control in Newborn Monkeys Immunized with an Influenza Virus HA Stem Nanoparticle” Crofts K.F et al. examined immunogenicity of H1ssF nanoparticle vaccine with AddaVax adjuvant in NHP model. The authors first tested H1ssF+AddaVax vaccine in adult monkeys to confirm H1ssF vaccine’s immunogenicity and protective efficacy. Although they saw good anti-stem binding antibody in H1ssF+AddaVax group but they didn’t show neither neutralizing antibody nor cellular immune responses after two i.m. immunizations. Then the authors tested H1ssF vaccine in newborn monkeys. They observed good binding antibodies against HA stem with some difference in quality.

Interestingly, the authors found that neutralizing Abs and ADCP activity are important contributors to Ab-mediated influenza virus control in newborn monkeys. The overall study is well performed, and conclusions are supported by results. Below are my concerns and suggestions.

We thank the reviewer for their positive comments regarding the study.

Major points:

- 1. What would make this work even more compelling is if anti-stem antibodies were given before neonatal immunization to mimic a setting more faithfully representing a real-life.** We agree that the impact of maternal antibody is an important question. To replicate the human situation, a complex array of influenza-specific Ab would need to be delivered to newborns, ideally through placental transfer. Such a study is beyond the scope of this manuscript which sought to understand the ability of this vaccine to elicit responses in the newborn and to uncover protective attributes. With that said we anticipate that the H1ssF+AddaVax vaccine may be particularly effective in the presence of maternal influenza antibodies based on the lower representation of stem-specific Abs in humans. As such, inhibition by maternal Abs is likely reduced compared to a head containing vaccine. Understanding the role of maternal antibody in infant influenza vaccine responses is a topic of future study for which the newborn AGM model will be of particular value. We have added a section about maternal Ab to the introduction **lines 65-75** and **lines 508-521** of the Discussion.
- 2. Figs. 1C & 3C symbol legends: Please remove symbols as they are confusing. I was looking for red triangles and circles. Just show colors in line will be more appropriate. Also please change one red square series to red circles in Fig. 1C as there are two red square lines.** We apologize for the duplicate symbols. We have made the suggested changes and appreciate the suggestion for increasing clarity of the figures.
- 3. p.7 line 4: “robust generation of stem-specific IgG at d10 p.v. in vaccinated adults”: Have the authors seen similar pattern in mice or other models? Could be the sensitivity issue of anti-monkey IgM antibody used?** In the absence of a monoclonal AGM IgM standard, we acknowledge that we may not be fully capturing the IgM signal. Secondary reagents can differ in their avidity and detection of subclasses. For this reason, we do not make any direct quantity comparisons between the IgM and total IgG at d10 p.v. time point. In reference to other animal models, there is limited data comparing IgM and IgG with this vaccine platform. We have performed a study (unpublished) in our lab where adult C57BL/6 mice were vaccinated with 3 µg of H1ssF+AddaVax. At d14p.v. (the closest time point we have to our study). The IgG:IgM ratio was greater than the adult AGM in our manuscript (mice=64, AGM=16). Thus, we believe the increased value for the IgG vs. IgM threshold titer in the AGM model is less likely to be due to a reagent limitation although we acknowledge this has not been formally demonstrated.
- 4. Please discuss about the lack of a defined regulatory pathway for approval of non-HA influenza vaccines.** This is clearly an important question for these vaccines. While these platforms may require additional defined immune correlates of protection, it is possible to move promising candidates that do not induce HAI antibodies forward. Multiple clinical trials of such vaccine candidates have been carried out or are ongoing by us and others and thus regulatory bodies are supportive of these vaccines. Currently, nothing is out of consideration by the regulatory agencies as a potential next generation influenza vaccine, including candidates based on HA stem, NA, NP, and M2 (all of which do not induce HAI antibody responses).
- 5. p. 25 line 17: This group might have anti-AGM IgG subclass antibodies. ‘Lara A. et al., Peripheral immune response in the African green monkey model following Nipah-Malaysia virus exposure by intermediate-size particle aerosol. PLoS Negl Trop Dis. 2019 Jun; 13(6):**

e0007454. doi: 10.1371/journal.pntd.0007454. We thank the reviewer for this suggestion. While the referenced paper utilizes AGM samples, the antibody subclasses are detected with antibodies developed for human IgG subclasses. None have been verified for AGM; thus, it is unclear how findings with these reagents translate to AGM IgG subclasses. In AGMs, two subclasses have been identified by sequencing- IgG1 and IgG4 and the Fc regions have greater similarity to macaque compared to human. IgG subclass reagents for macaque IgG1 and IgG2 have been developed. We obtained these antibodies to test in our AGM model. Unfortunately, we observed no binding to AGM Ab. We also used a panel of human specific IgG subclass Abs to look for any differences in binding patterns in plasma derived AGM Ab; however, we again saw no binding. These experiments highlight the importance of species-specific Fc reagents that will allow identification of IgG subclass.

- 6. Can the authors add comments about safety of H1ssF+AddVax vaccine in neonates?** In addition to the data in SF4A that show similar CRP levels in the H1ssF+AddaVax animals compared to the ctrl/H1ssF animals, we include each infant's temperature at ~24h p.v. and weight over time (**lines 238-240**, Fig. S4C). No significant difference in weight at any of the time points was observed among the different vaccine groups. Further, following vaccination, infants are monitored for 15 mins prior to return to their mothers. No immediate reaction was noted. The infant and mother pair are subsequently housed in a single cage for 24h before return to the colony. At the 24h p.v. blood draw, infants are again assessed for changes at the vaccination site as well as general health characteristics (i.e. alert, latching on to mothers). No vaccine-induced changes were observed.

Minor points:

- 1. "ELISA for the Detection of PR8-Specific IgG" section; "14 µg/well" should be "14 µg/mL". Also please confirm the OD. If you don't stop TMB reaction OD should be 650nm.**

The PR8 concentration has been corrected (**line 34**, Supplemental information). ELISA plates are stopped with 25 µl of 2N H₂SO₄. This is included in the Materials and Methods and additional sites moving forward in this section.

- 2. Please ensure correct formatting (sub/superscript) of CO₂, 10⁵ cells, IC₈₀ etc.** This has been corrected.

- 3. Please ensure consistent description for "AddaVax". AddaVax, Addavax etc.**

We apologize for the lack of consistency. This has been corrected.

- 4. Please ensure correct description for "A/California/07/2009" not "A/California/07/09".**

This has been corrected.

- 5. p.16 line 8: There is no Supplementary Fig. 5a. Should be Fig. 5?**

This has been corrected.

- 6. Fig. 4F: "PBS" should be "Ctrl" for consistency.** We have made the suggested change.

Reviewer #3

Overall comments: The authors tested an HA stem based vaccine with Addavax (an MF59-like adjuvant) in juvenile african green monkeys for breadth of antibody responses and reduction of viral titers in BALs. The vaccine elicited antibodies that neutralized an H1N1 and H5N1 historical viruses. Overall, the study is well performed. Several studies have used HAstem vaccines, but this is the first study to examine vaccine effectiveness in young monkeys. To determine the effectiveness of antibodies elicited by HAstem, the study would be stronger if the antibodies were tested against

modern group 1 influenza vaccines, i.e. 2023-2024 season to show that the elicited antibodies protect against drifted H1N1 variants.

We thank the reviewer for suggesting this experiment. We have now evaluated the ability of vaccine-elicited IgG to bind the 2023-2024 Fluzone vaccine. We find significant binding by Abs from H1ssF+AddaVax vaccinated animals compared to the H1ssF and the Ctrl animals. The level of Fluzone binding and NC99 stem binding were highly correlated ($r = 0.81$, $r^2 = 0.66$, p value = 0.015). These data support the ability of the stem-specific IgG to recognize modern group 1 strains. These data are included in Fig. S6C and D and are discussed in **lines 288-298** in the manuscript.

We appreciate the time of the reviewers in re-evaluating our manuscript. Please see below for the response to Reviewer #1. Reviewer #2 had no further comments.

Reviewer #1 (Remarks to the Author):

Overall, the authors performed a comprehensive response and editing to the reviewers comments, which has improved the quality and translational potential of the findings.

We appreciate this comment.

My only recommendation is to include the injection site observations, even though negative, to the results of the manuscript and to add the limitation of the AGM model for studying B cell repertoires in the discussion.

Both of these have been added.